# Regulation of proteostasis by sleep through autophagy in *Drosophila* models of Alzheimer's disease

Natalie Ortiz-Vega[1,2,3] , Amanda G Lobato[1] , Tijana Canic[2,4] , Yi Zhu[2] , Stanislav Lazopulo[4] , Sheyum Syed[4] , R Grace Zhai[1,2]

Sleep and circadian rhythm dysfunctions are common clinical features of Alzheimer's disease (AD). Increasing evidence suggests that in addition to being a symptom, sleep disturbances can also drive the progression of neurodegeneration. Protein aggregation is a pathological hallmark of AD; however, the molecular pathways behind how sleep affects protein homeostasis remain elusive. Here we demonstrate that sleep modulation influences proteostasis and the progression of neurodegeneration in *Drosophila* models of tauopathy. We show that sleep deprivation enhanced Tau aggregational toxicity resulting in exacerbated synaptic degeneration. In contrast, sleep induction using gaboxadol led to reduced toxic Tau accumulation in neurons as a result of modulated autophagic flux and enhanced clearance of ubiquitinated Tau, suggesting altered protein processing and clearance that resulted in improved synaptic integrity and function. These findings highlight the complex relationship between sleep and regulation of protein homeostasis and the neuroprotective potential of sleep-enhancing therapeutics to slow the progression or delay the onset of neurodegeneration.

## Introduction

Alzheimer's disease (AD) is the most common neurodegenerative disorder and the leading cause of dementia (1). Affected individuals display a variety of progressive and disabling neurological symptoms such as cognitive decline, motor dysfunction, and psychiatric symptoms (2). The main pathological phenotype of AD is shrinkage of the hippocampus and cortex, the accumulation of plaques formed by extracellular insoluble aggregates of amyloid-beta fragments (Aβ), and neurofibrillary tangles (NFTs) formed by intracellular accumulation of hyperphosphorylated Tau (pTau) (3, 4).

Tau proteins are microtubule-binding proteins that promote the stabilization and assembly of the microtubules. Under pathological conditions, Tau is hyperphosphorylated, detached from microtubules, and forms neurotoxic oligomers that eventually aggregate into paired helical fragments (PHFs) and eventually form NFTs (5). It has been found in AD brains that Tau is three to fourfold more hyperphosphorylated than in normal adults, causing disruption of its normal function and facilitating its polymerization into PHFs and forming NFTs (6). Phosphorylation of Ser202/Thr205 is used to assign a Braak stage, a method to classify the severity of pathology in AD, based on the presence of PHF-Tau. At Braak stage V/VI, phosphorylation at this site is 4-fold to 13-fold above control levels (7). Phosphorylation site S262 is inside the microtubule-binding region and has been shown to promote Tau oligomerization and to be critical to mediate toxicity (8, 9).

Sleep disturbances are observed in ~45% of patients and may be one of the first manifestations of the disease (2, 10, 11). These sleep disorders include insomnia, excessive daytime sleepiness, and fragmented sleep (2). Specifically, polysomnography studies show reduced total sleep time and sleep efficiency, increased sleep-onset latency, and wake time after sleep onset, reduced deep sleep an REM sleep amounts (2, 12). This is particularly important because increasing evidence suggests a bidirectional relationship between sleep and neurodegenerative diseases, where on one side, neurodegeneration can lead to circadian dysregulation and sleep disorders (13, 14), whereas on the other hand, sleep disruption can promote neurodegeneration through induced neuroinflammation, and aberrant protein homeostasis (15). Meta-analysis results suggest that individuals with disturbed sleep have a 1.68 times higher risk of developing cognitive impairment and/or AD (16). Moreover, sleep deprivation is shown to affect the glymphatic clearance of proteins in the brain and increases the accumulation of Aβ in the hippocampal region, and the accumulation of hyperphosphorylated Tau leading to inhibition of neurogenesis and cognitive dysfunction (16, 17, 18, 19).

[1]Department of Neurology, University of Chicago, Chicago, IL, USA   [2]Department of Molecular and Cellular Pharmacology, University of Miami Miller School of Medicine, Miami, FL, USA   [3]Graduate Program in Molecular and Cellular Pharmacology, University of Miami Miller School of Medicine, Miami, FL, USA   [4]Department of Physics, University of Miami, Coral Gables, FL, USA

Correspondence: rgzhai@uchicago.edu
Yi Zhu's present address is Department of Laboratory Medicine and Pathology, Mayo Clinic, Rochester, MN, USA
Stanislav Lazopulo's present address is Department of Physics and Center for Brain Science, Harvard University, Cambridge, MA, USA

Recent studies suggest that sleep regulates autophagy in a daily sleep/wake cycle, and sustained changes in autophagosome level affect sleep amount (20). Although sleep disorders have been linked to multiple neurodegenerative diseases, few studies have fully investigated how sleep affects AD progression. Previous studies have indicated that sleep abnormalities exacerbate neurodegeneration and that sleep therapies may slow down disease progression. For example, in a transgenic mouse model of Aβ deposition, it has been observed that sleep deprivation accelerates plaque accumulation, whereas promoting sleep with an orexin antagonist significantly inhibits plaque formation (7). However, the molecular mechanisms of how these interventions lead to pathophysiological changes in AD remain unclear.

Sleep abnormalities are highly present in AD patients before the clinical onset of the disease (8), therefore, uncovering the sleep-neurodegeneration relationship longitudinally and before disease manifestation is required. *Drosophila* is a powerful genetic model system to study AD and sleep (9). Several models have been characterized to model given its complex etiology (21, 22). Specifically, for this study some *Drosophila* models for tauopathy have been established by expressing either WT (Tau$^{WT}$) or mutant (Tau$^{R406W}$) human Tau protein in neurons. Using the GAL4/UAS system, a transgene of interest, in this case, Tau$^{WT}$ or Tau$^{R406W}$, can be expressed in specific tissues. Expressing either Tau$^{WT}$ or Tau$^{R406W}$ in all neurons, using *elav-GAL4* or in photoreceptors using *GMR-Gal4*, recapitulate several salient features of AD pathology including the adult-onset, progressive neurodegeneration with altered lifespan and accumulation of hyperphosphorylated Tau (pTau) (8, 23). Tau toxicity and dysfunction have been widely studied using different *Drosophila* drivers and phenotypic readouts. Some common readouts include eye morphology, brain vacuolization, lifespan, lethality, axonal transport, and loss of olfactory learning and memory have been used to measure Tau neurotoxicity (24). In this report, we took advantage of these well-characterized *Drosophila* models of tauopathy to address the molecular mechanisms of how sleep abnormalities affect proteostasis and Tau-induced neurodegeneration in vivo.

## Results

### Establish a sleep modulation paradigm to assess altered progression of neurodegeneration

Similar to humans, *Drosophila* follows a 12-h day/12-h night rhythm (25). Sleep in flies is measured as 5 min or more of inactivity (26, 27). To study *Drosophila* locomotor activity and sleep we used a *Drosophila* Activity Monitor (TriKinetics) using a locally built detection system that includes an infrared (IR) beam to record the locomotor activity of solitary flies placed in glass tubes (Fig S1A and B), as commonly used in measuring fly sleep (28). Using the activity monitoring system, we first determined the sleep patterns of tauopathy models. *Drosophila* models for tauopathy, established by expressing either wild-type (Tau$^{WT}$) or mutant (Tau$^{R406W}$) human Tau protein in neurons, recapitulate adult-onset, progressive neurodegeneration with altered lifespan, and accumulation of pTau (29). We expressed either *UAS-CD8-GFP*, *UAS-Tau$^{WT}$*, or *UAS-*

Tau$^{R406W}$ pan-neuronally using *elav-GAL4* driver or in the photoreceptors using *GMR-GAL4* driver as previously reported (29, 30), and monitored *Drosophila* activity behavior for 7 d, which included 2 d of acclimation and 4 d of sleep monitoring (Fig 1A–F). This experiment was designed to examine the early effects of sleep modulation before the onset of severe neurodegeneration in tauopathy models (30). Collectively, combining pan-neuronal (*elav-GAL4*) or cell-type specific expression, in this case, photoreceptor expression of Tau (*GMR-GAL4*) models with successful sleep modulation allows the mechanistic dissection of the impact of sleep modulation on neurodegeneration in *Drosophila* models of tauopathy.

Pan-neuronal expression of Tau$^{WT}$ or Tau$^{R406W}$ with *elav-GAL4* caused significant sleep behavior changes (Fig 1A), specifically in reduced average sleep time per 24 h of 117 min less in Tau$^{WT}$, and 197 min less in Tau$^{R406W}$ (Fig 1B); increased number of sleep bouts (Fig 1C), and decreased length of each sleep bout (Fig 1D). Combined, these results show increased sleep fragmentation and decreased total sleep length, suggesting that Tau$^{WT}$ and Tau$^{R406W}$ expression in all neurons causes sleep disturbances, consistent with previous reports that have observed sleep disturbances caused by pan-neuronal expression of Aβ or Tau (31, 32, 33). Such significant change in sleep behavior makes it challenging to dissect the direct effects of sleep modulation on neurodegeneration. To focus on the effects of sleep modulation on neurodegeneration, and minimize the influence of Tau expression on sleep disturbance, we incorporated an additional tauopathy model, by expressing Tau in a subset of neurons with minimum impact on sleep behavior. To that end, we analyzed whether expressing Tau in the photoreceptors using *GMR-GAL4* caused sleep disturbances by monitoring their sleep behavior (Fig 1E). When comparing Tau$^{WT}$ or Tau$^{R406W}$ to controls (GFP), there was no significant difference in either the amount of average sleep per 24 h (Fig 1F), the number of sleep bouts (Fig 1G), or the average length of each sleep bout (Fig 1H). Detailed analyses of sleep divided into night and day showed that expression of Tau in the photoreceptors had minimum effects on their innate sleep behavior (Fig S2A–C) and no significant change in number of sleep bouts (Fig S2C) or bout length (Fig S2D). Together, these data suggest that expression of Tau in the photoreceptors causes minimum effect on sleep, making it a feasible model for us to examine the molecular impact of sleep modulation on Tau aggregation in a system with relatively normal sleep behavior. Moreover, in *Drosophila*, gaboxadol induces sleep through the GABA$_A$ receptors, specifically, *Ligand-gated chloride channel homolog 3 (Lcch3)* and the *GABA and glycine-like receptor (Grd)* (34). Previous research has shown these receptors are present in interneurons postsynaptic to photoreceptors (35, 36). Therefore, incorporating the expression of Tau in the photoreceptors, *using GMR-GAL4 as a driver*, also allows the confirmation that any synaptic effects observed when feeding gaboxadol are because of increased sleep and not caused by direct activation of GABA receptors.

To establish a sleep modulation paradigm that induces significant changes in sleep time, we incorporated both sleep deprivation and sleep induction protocols. Sleep disruption was achieved using a mechanical "deprivator" that rotates 90° for a determined time (37). During mechanical deprivation, we can

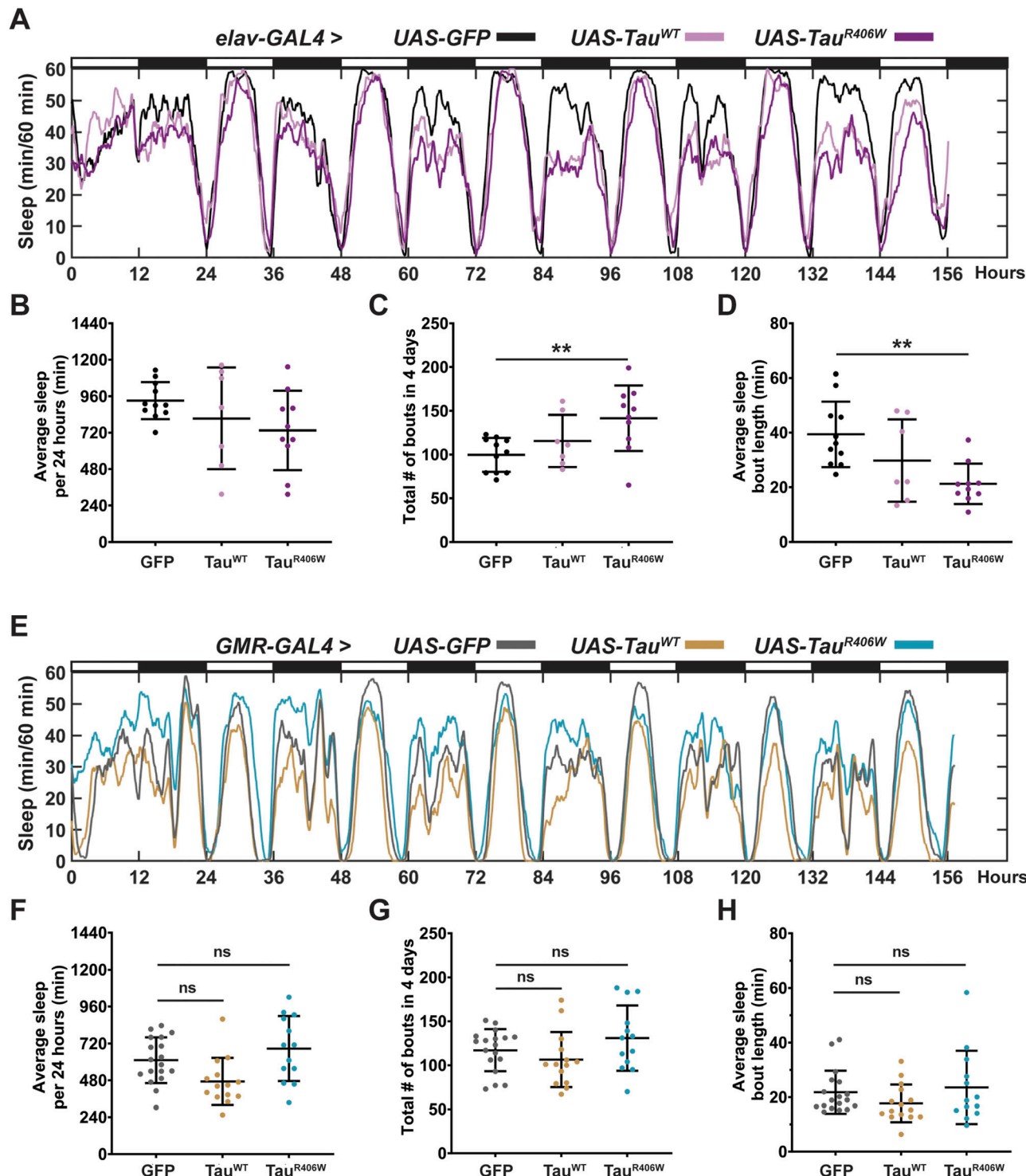

**Figure 1. Pan-neuronal Tau expression alters sleep frequency and length, whereas photoreceptor expression causes minimal effects on sleep.**
**(A)** Sleep profiles of 2 DAE flies expressing either *UAS-CD8-GFP* (black), *UAS-hTau^WT^* (light pink), or *UAS-hTau^R406W^* (magenta) pan-neuronally using *elav-GAL4* driver. Flies were allowed to acclimate from 0–48 h and sleep was measured from 48–144 h. **(B)** Quantification of average sleep per 24 h measured in minutes. **(C)** Quantification of total number of bouts in 4 d (40–144 h). **(D)** Quantification of average sleep bout length in minutes. **(E)** Sleep profiles of 2 DAE flies expressing either *UAS-CD8-GFP* (gray), *UAS-hTau^WT^* (gold), or *UAS-hTau^R406W^* (turquoise) in the photoreceptors using *GMR-GAL4* driver. **(F)** Quantification of average sleep per 24 h measured in minutes. **(G)** Quantification of total number of bouts in 4 d. **(H)** Quantification of average sleep bout length in minutes. Data as mean ± SD. n = 7–22, one-way ANOVA. **$P < 0.01$.

observe significant reduction in sleep, followed by an earlier sleep onset and increased daytime sleep. These changes in sleep profile are because of the sleep homeostat attempting to reduce the amount of sleep need accumulated during a deprivation event. Sleep induction was achieved by gaboxadol feeding following previously published methods (20). Flies were subjected to either mechanical sleep deprivation for two nights, 9 h each night (hours 111–120 and hours 135–140) or gaboxadol feeding at 0.1 mg/ml concentration for 4 d (hours 48–156) (Fig 2A). Sleep studies showed significant sleep behavior alterations after sleep deprivation or sleep induction in all groups (Fig 2B). On average, sleep deprivation resulted in a reduction in total nighttime sleep by 160, 160, and 95 min for GFP, Tau$^{WT}$, and Tau$^{R406W}$, respectively; whereas sleep induction resulted in an increase in nighttime sleep of 174, 185, and 145 min, respectively. Daytime activity showed similar changes by sleep deprivation or induction (Fig 2C). Detailed sleep bout analyses showed that the number of sleep bouts was decreased by sleep deprivation, when increased by sleep induction (Fig 2D), without significant changes in sleep bout length (Fig 2E). Taken together, these results showed successful sleep modulation after mechanical deprivation or sleep induction for all groups and demonstrated the feasibility of the optimized sleep modulation paradigm.

It is important to note that all experiments were started using adult flies of 2 DAE (days after eclosion) and cellular and molecular analyses were performed 4 h after the conclusion of sleep modulation (160 h) on 8 DAE (Fig 2A). This paradigm was designed to examine the early effects of sleep modulation before the onset of severe neurodegeneration in tauopathy models (30).

## Sleep modulation alters the progression of Tau-induced synaptic degeneration

Synaptic loss is one of the early events of neurodegeneration and main lesions presented in human AD patients (38). We have previously reported that our Tau expression models recapitulate the phenotype of Tau-induced synaptic degeneration (29). To evaluate how sleep modulation affects synaptic integrity and function we expressed Tau$^{WT}$ or Tau$^{R406W}$ in the *Drosophila* visual system to take advantage of the highly organized parallel axons of the compound eye: the R1–R6 photoreceptors axons cross the lamina cortex and make synaptic connections at the lamina neuropil, whereas R7–R8 extended their axons beyond the lamina and project to the medullar neuropil (39, 40). The lamina synapses can be visualized in the x-y and x-z planes (Fig 3A). *Drosophila* lamina synapses are organized into repetitive structures called "cartridges" reflecting the compound eye organization. Each cartridge contains an assemble of synapses formed between six photoreceptor neurons (one of each R1-R6) and the lamina monopolar cells (39). Analyzing synaptic morphology at the x-y and x-z planes provides a three-dimensional view of the synaptic structure. Using Bruchpilot (BRP), an endogenous active zone-associated cytomatrix protein as a marker, we analyzed synaptic structure under different sleep modulation conditions with confocal imaging (Fig 3B).

The synaptic architectural integrity as indicated by tissue thickness of the lamina neuropil was altered by sleep modulation. Whereas sleep modulation had no significant effect on tissue

thickness in normal control (GFP) group, sleep deprivation caused an ~11.5% reduction, and sleep induction an ~9.5% increase in tissue thickness of Tau$^{WT}$. In Tau$^{R406W}$, sleep deprivation caused a ~4.6% reduction, whereas sleep induction showed a ~5.7% increase in tissue thickness, on average (Fig 3C).

Measurement of total BRP intensity of the lamina neuropil in the x-z plane, in the normal control (GFP) group showed no change after sleep deprivation but a ~56% increase after sleep induction, suggesting a physiological effect of sleep induction on active zone BRP homeostasis. In tauopathy groups (Tau$^{WT}$ and Tau$^{R406W}$), the BRP level was reduced by 32% after sleep deprivation and increased by 46–55% after sleep induction (Fig 3D). Moreover, we quantified the average BRP intensity per lamina cartridge and found a Tau expression-induced, significant disruption of cartridge integrity and reduced BRP intensity per lamina cartridge. This reduction was exacerbated with sleep deprivation and significantly improved after sleep induction for both Tau$^{WT}$ and Tau$^{R406W}$ (Fig 3E). Taken together, these results suggest sleep deprivation exacerbated Tau-induced loss of active zone structures in the presynaptic terminals, whereas sleep induction significantly improved the localization of endogenous BRP at the synapses.

To further extend the analysis of the effects of sleep modulation on synaptic integrity, we analyzed the synaptic localization of endogenous cysteine-string protein (CSP), a synaptic vesicle-associated chaperone critical for neurotransmission (41) (Fig S3A). Quantification of CSP intensity per lamina cartridge revealed a significant reduction in CSP per lamina cartridge in both Tau$^{WT}$ and Tau$^{R406W}$ groups. Furthermore, after sleep deprivation, a significant reduction in CSP was observed in the Tau$^{WT}$ group. Lastly, a significant increase was observed after sleep induction in the Tau$^{R406W}$ group (Fig S3B). Consistent with BRP levels, these results suggest that sleep deprivation exacerbated overall synaptic loss and integrity and sleep induction provided significant improvement of synaptic structures.

To examine the functional consequence of sleep modulation, we carried out electroretinogram (ERG) study of synaptic function. These electrical responses caused by light stimulation provide information about phototransduction (receptor potential, RP) and pre- and postsynaptic responses of the laminar monopolar cells (On and Off transients) (23, 24, 42). Consistent with previous reports, tauopathy flies showed reduced On and Off transients and receptor potential under control conditions, indicating synaptic degeneration (43). Notably, these defects were partially ameliorated by sleep induction, which showed significant increases in the amplitude of On and Off transients, in all groups (Fig 4A–D). On the other hand, sleep deprivation showed no significant ERG changes. The minimal change caused by sleep deprivation might indicate that a stronger or more prolonged deprivation effect is required to observe substantial functional degeneration at this early stage of the disease. The improvements in On and Off transients after sleep induction suggest improved synaptic function, which is consistent with the overall improvement in synaptic morphology that was previously described. Similarly, deprivation did not result in RP amplitude deterioration, but sleep induction showed a slight RP amplitude increase in normal control GFP flies (Fig 4D), indicating that sleep induction has a general effect on all synapses.

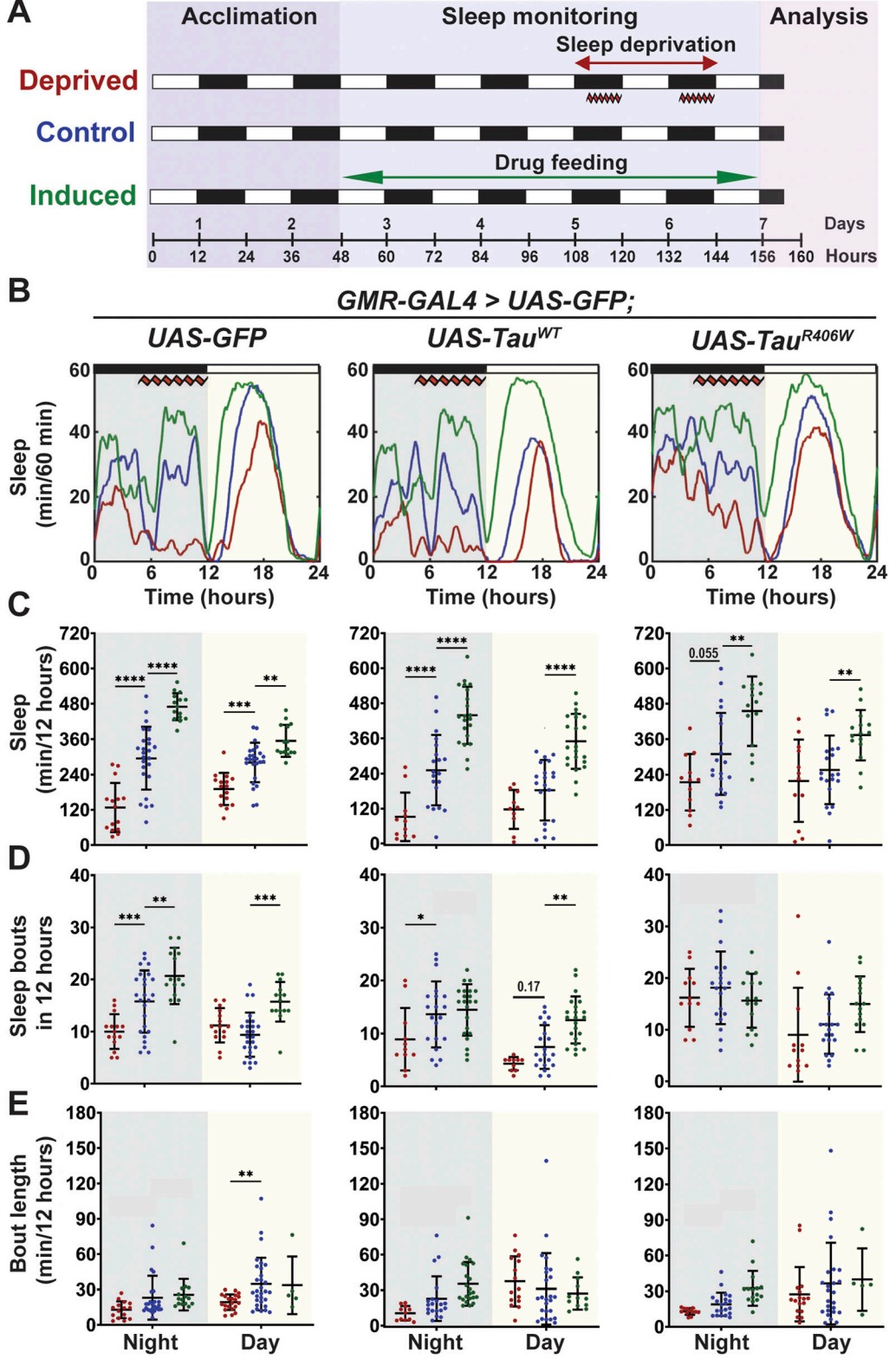

**Figure 2. Established sleep paradigm successfully modulates total sleep time.**
**(A)** Sleep modulation paradigm. 2 DAE flies were placed in single glass tubes with food and their locomotor activity was monitored for 7 d. Deprived group was mechanically deprived for 9 h on the last two nights of the experiment (hours 111–120 and hours 135–144). The induced group was placed on food containing 0.1 mg/ml gaboxadol for 4 d (hours 48–144). Cellular and molecular analyses were performed at 160 h. **(B)** Flies were expressing either *UAS-CD8-GFP, UAS-hTau^WT*, or *UAS-hTau^R406W* in the photoreceptors using *GMR-GAL4* driver. Sleep per 60 min traces showing control (blue), deprived (red), induced (green) groups. **(C)** Total sleep time per 12 h during night (gray box) and day (yellow box) for hours 132–156 is quantified. **(D)** Number of sleep bouts per 12 h. **(E)** Sleep bout length in minutes per 12 h. Data as mean ± SD. n = 10–25, one-way ANOVA. *P < 0.05, **P < 0.01, ***P < 0.001, ****P < 0.0001.

Taken together, our morphological and functional study of the synapse demonstrates that sleep modulation is altering the progression of Tau-induced synaptic degeneration. Specifically, sleep deprivation accelerates synaptic degeneration as shown by exacerbated loss of BRP and CSP and overall synaptic structural integrity, whereas sleep induction improves overall synaptic physiology and structural integrity.

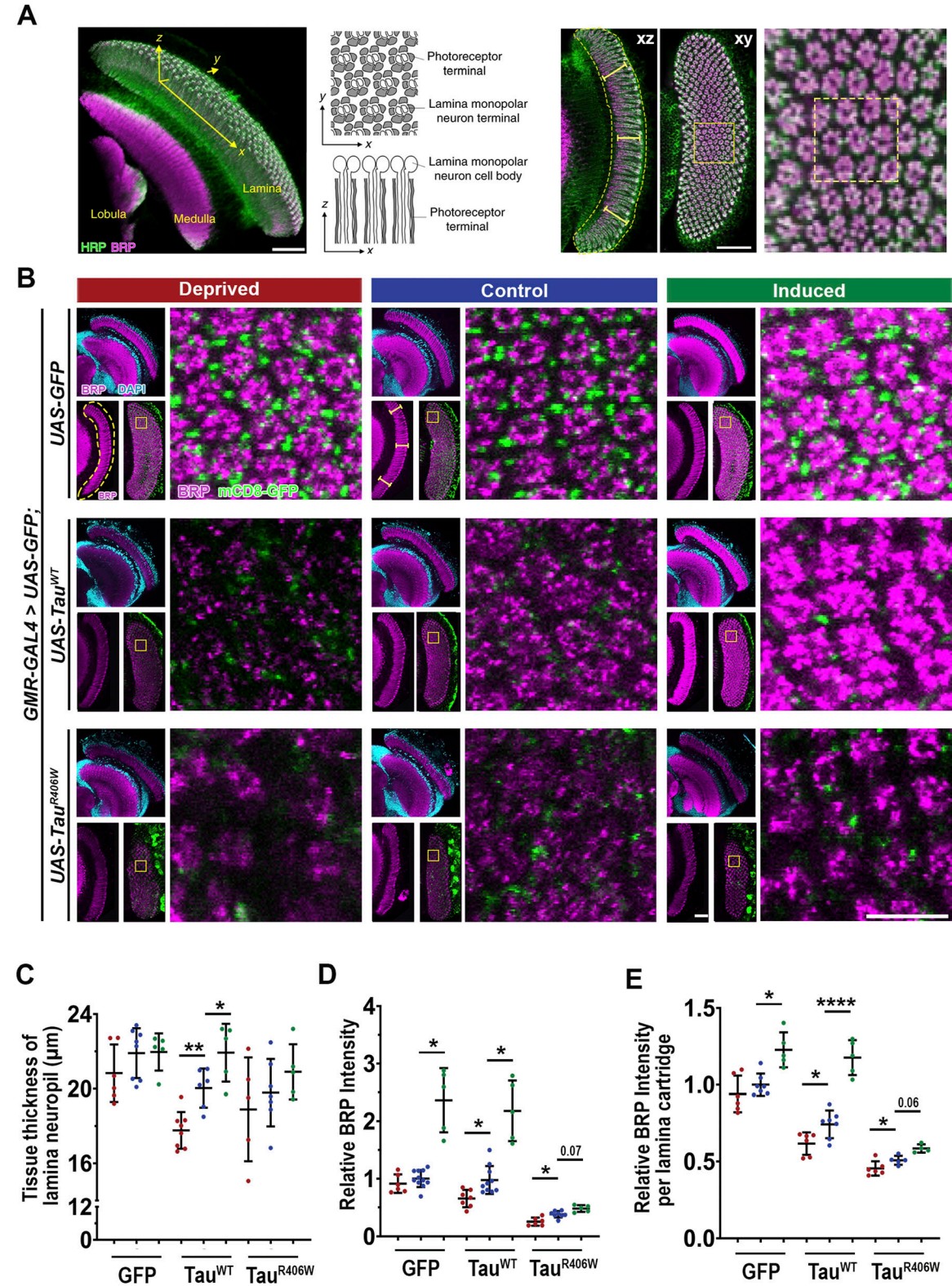

**Figure 3. Sleep modulation influences Tau-induced impaired synaptic integrity and morphology.**
**(A)** The three-dimensional structure of the *Drosophila* visual system showing the lamina, medulla, and lobula. The x-z and x-y planes showing the photoreceptor terminals and lamina neurons are indicated. The organized lamina cartridges, including a higher magnification example and columnar photoreceptor neurons, are shown in the x-z and x-y planes, respectively. Yellow-dashed box shows 3 × 3 example of lamina cartridges used for quantification of x-y plane. **(B)** Lamina structures at 8 DAE containing endogenous mCD8-GFP (green) probed for BRP (magenta) and stained with DAPI (cyan). Yellow-dashed line highlights lamina neuropil in x-z plane.

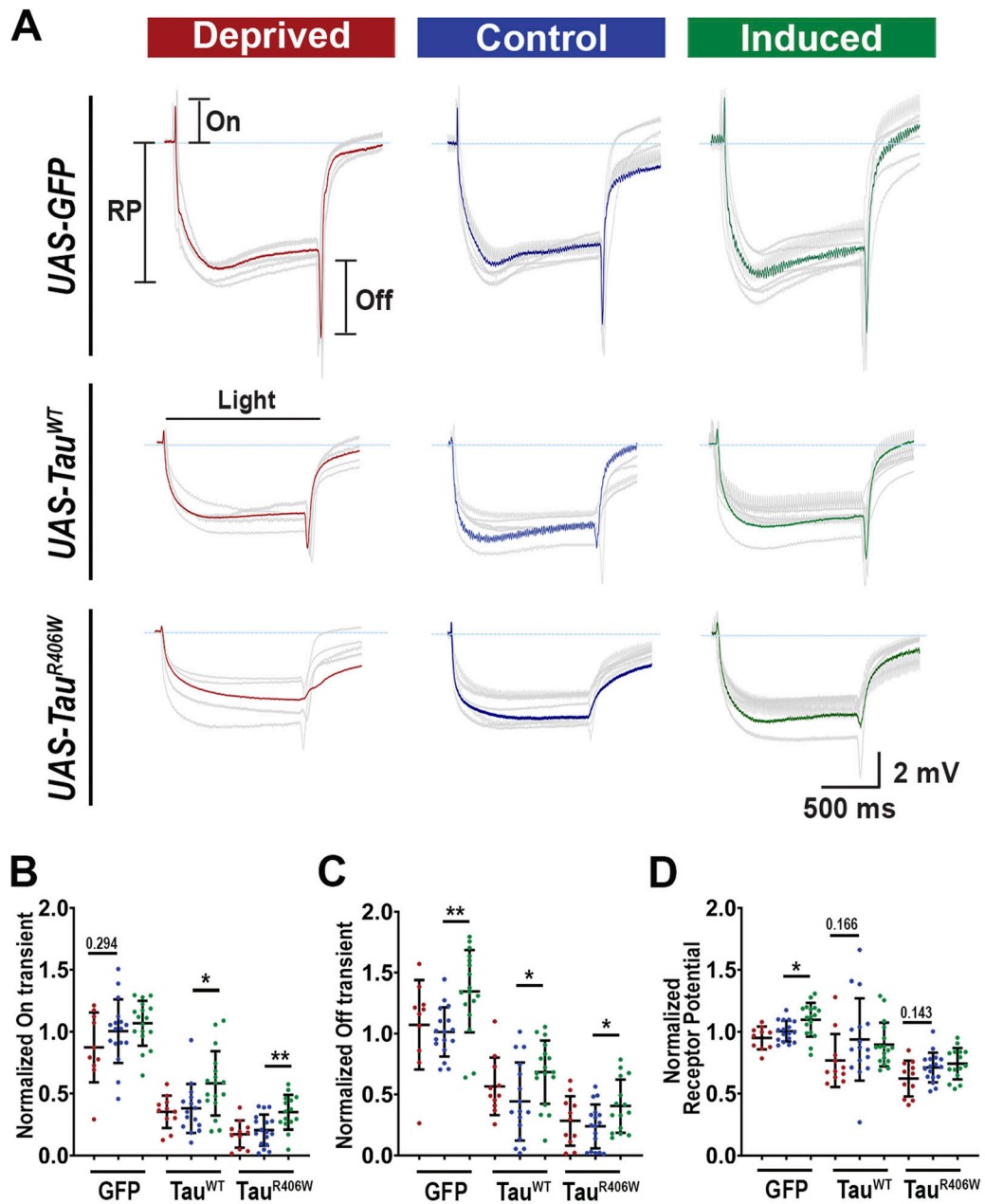

**Figure 4. Sleep modulation alters Tau-induced synaptic dysfunction.**
**(A)** Electroretinogram recordings showing the average response of 10 flies (gray). Average is shown in red, blue, and green, respectively. RP: resting potential; On, On transient; Off, Off transient. **(B, C, D)** Quantification of transients normalized to the GFP control. On transient (B), Off transient (C), and receptor potential (D) amplitude. For ERGs, n ≥ 10 light pulses recorded from 10–12 flies for each group. Data are presented as mean ± SD. One-way ANOVA. *$P < 0.05$, **$P < 0.01$.

## Sleep modulation affects hyperphosphorylated Tau (pTau) accumulation in neurons

To dissect the molecular underpinnings of the effect of sleep modulation on protein homeostasis we first analyzed how sleep modulation affected general protein processing using the reporter GFP protein. In our tauopathy model, we co-expressed mCD8-GFP, a membrane-targeted GFP as a reporter to monitor the neuronal membrane integrity (Fig 5A). In the control group where only GFP was expressed, normal photoreceptor morphology is marked by GFP fluorescence, showing the neuronal membrane of photoreceptor neurons R1–6, which extend into lamina neuropil, and R7–R8

Yellow-dashed box highlights organized lamina cartridges in x-y plane. Scale bar 10 µm. Cellular and functional analyses were performed at 160 h. **(B, C)** Quantification of tissue thickness of lamina neuropil (yellow bars shown on panel (B)). **(D)** Quantification of BRP intensity in the x-z plane of the lamina neuropil normalized to GFP control (yellow-dashed line). Brown-Forsythe and Welch ANOVA with Dunnett's multiple comparison correction. **(E)** Quantification of average BRP intensity per lamina cartridge in x-y plane (yellow-dashed box). For dissections, n = 5–11 for each group. Data are presented as mean ± SD. One-way ANOVA. *$P < 0.05$, **$P < 0.01$, ****$P < 0.0001$.

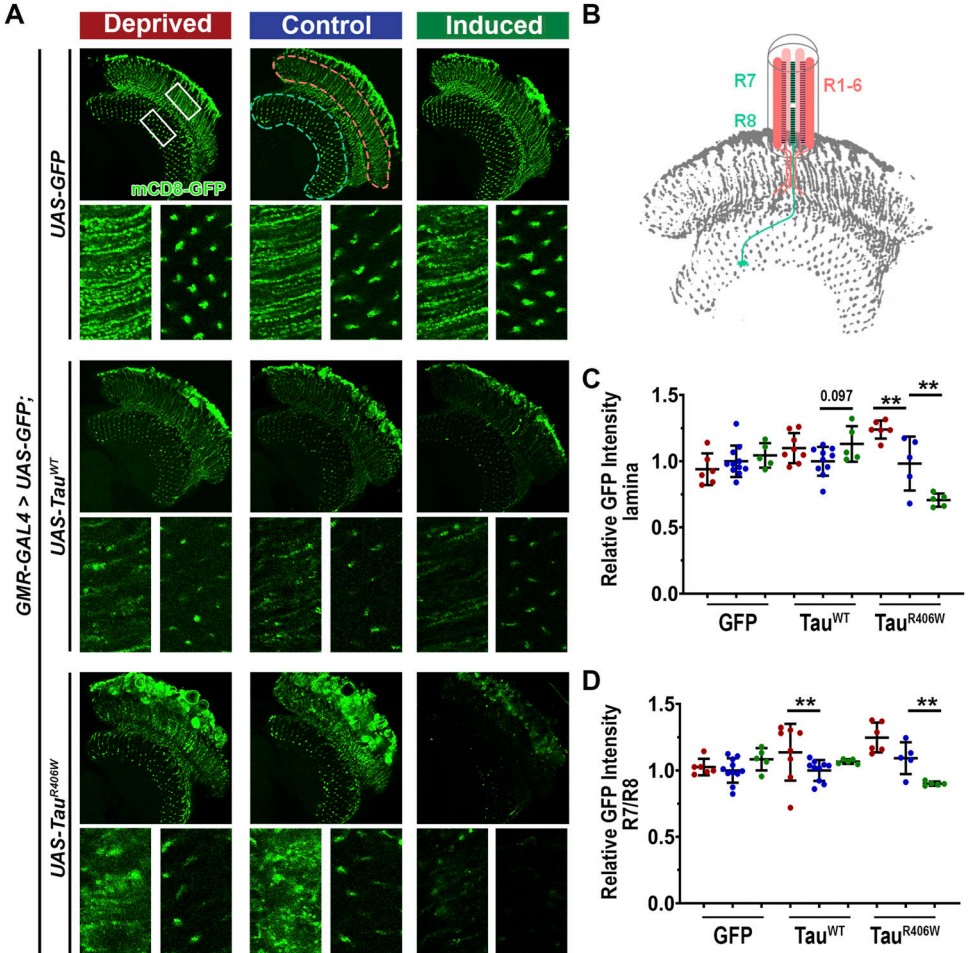

**Figure 5. Sleep modulation alters membrane-targeted GFP accumulation in neurons when co-expressed with Tau.**
**(A)** *Drosophila* optic lobe was scanned. White boxes represent higher magnification of lamina (left) and medulla (right). Scale bar 10 μm. **(B)** Pink-dashed line shows lamina neuropil showing photoreceptor synaptic terminals R1-6, whereas blue-dashed line shows synaptic terminals of photoreceptors R7/R8. **(C)** Intensity of GFP in lamina neuropil normalized to GFP control. One-way ANOVA. **(D)** Quantification of intensity of GFP in medulla neuropil. Data as mean ± SD. n = 5–11, Brown-Forsythe and Welch ANOVA with Dunnet's multiple comparison correction. *P < 0.05, **P < 0.01.

terminals which are observed in the medulla neuropil (Fig 5B). Importantly, the GFP intensity and localization pattern were largely unchanged by sleep modulation, suggesting the minimal effect of sleep modulation on protein homeostasis in normal healthy neurons. However, in Tau-expressing flies, after sleep deprivation there was a significant increase in GFP in the lamina for the Tau^R406W group, and a significant reduction after sleep induction (Fig 5C). In addition, quantification of the intensity at the R7/R8 terminals in the medulla neuropil shows a significant increase after sleep deprivation in the Tau^WT group and a significant reduction after sleep induction in the Tau^R406W group (Fig 5D). Taken together these results show that sleep modulation had minimal effects on reporter GFP homeostasis in normal conditions, whereas significantly alters protein homeostasis under pathological Tau-expressing conditions. The increase in reporter GFP accumulation after sleep deprivation could indicate a block in protein processing and toxic protein buildup in tauopathy conditions, whereas the decrease in reporter GFP after sleep induction could suggest enhanced clearance of misfolded GFP proteins and reduction in toxic protein burden.

To dissect the molecular underpinnings of the effect of sleep modulation on Tau-induced synaptic degeneration, we set out to examine the biochemical changes of Tau. Expression of Tau^R406W results in Tau hyperphosphorylation and synaptic aggregation of hyperphosphorylated pTau (30). We have optimized a high-resolution imaging approach that allows the characterization of the properties of pTau and the dynamic interaction with cellular organelles (30, 44). Using pTau-specific antibody AT8 (for Ser202, Thr205) that is specific for Tau pared helical fragments (PHF-tau), we quantitatively analyzed overall optic lobe pTau intensity and individual size distribution and intensity of pTau clusters to examine changes in the aggregational state of Tau caused by sleep modulation (Fig 6A). The total intensity of pTau showed a remarkable alteration under sleep modulation with a significant increase after sleep deprivation and a modest reduction after sleep induction in Tau^R406W (Fig 6B). When comparing individual pTau clusters a significant increase in intensity was observed after sleep deprivation for Tau^WT and a slight increase in Tau^R406W (Fig 6C). The size analysis of pTau cluster revealed an interesting effect of sleep modulation, where sleep deprivation significantly increased medium and large clusters in Tau^R406W and Tau^WT, respectively, whereas sleep induction decreased medium and large-sized clusters in Tau^WT (Fig 6D).

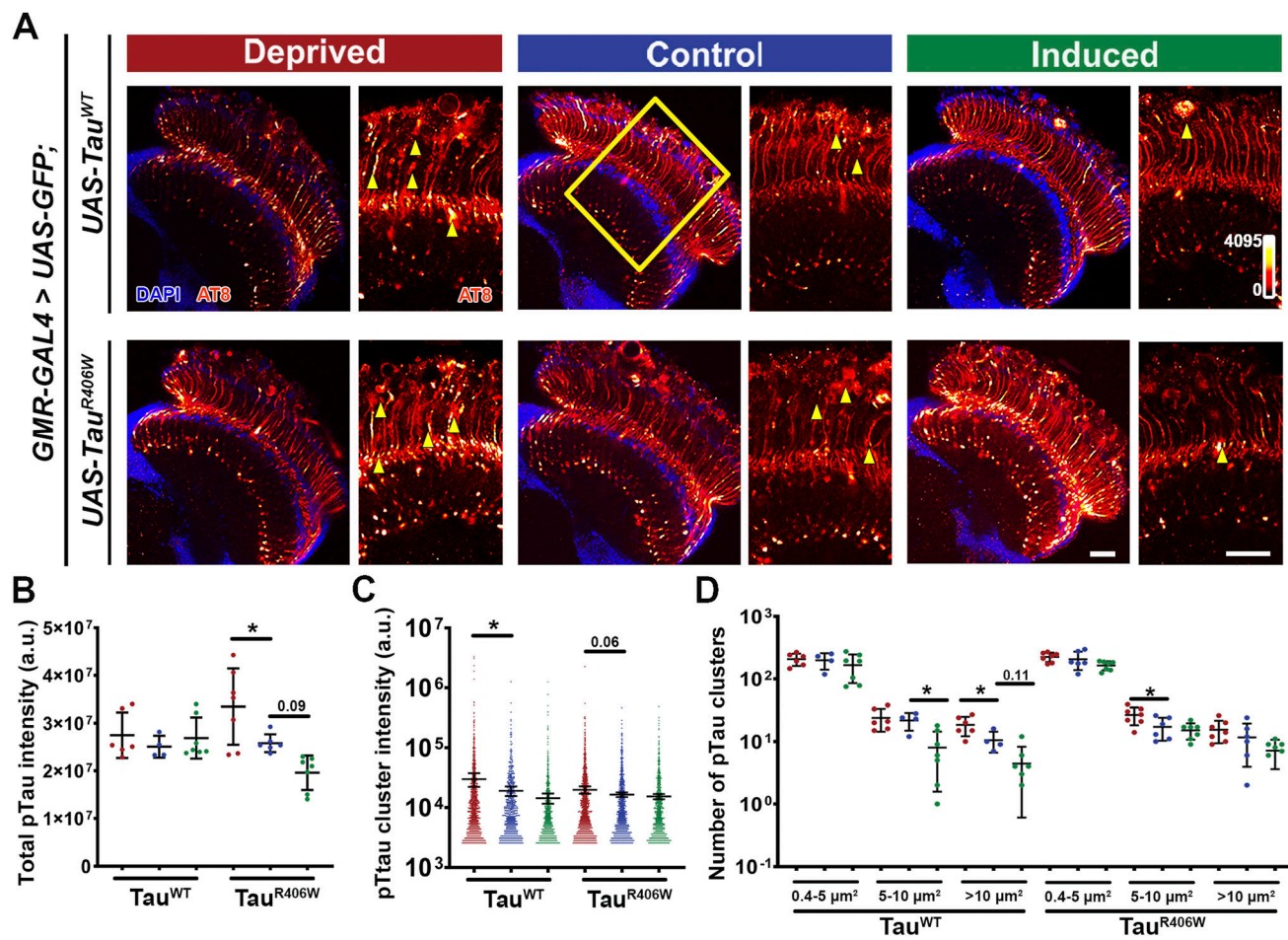

**Figure 6. Sleep modulation alters hyperphosphorylated Tau accumulation in neurons.**
**(A)** *Drosophila* optic lobe was stained with hyperphosphorylated Tau antibody AT8 (Ser[202], Thr[205]) (heatmap 0-4095). Yellow box represents zoomed-in region of interest used for quantification. Scale bar 30 $\mu m$. **(C, D)** Yellow arrowheads represent AT8 clusters quantified in panels (C, D). **(B)** Quantification of total AT8 intensity of optic lobe. **(C)** Quantification of intensity of AT8 clusters. **(D)** Quantification of number of AT8 clusters divided in small (0.4 < 5 $\mu m^2$), medium (5–10 $\mu m^2$), and large (>10 $\mu m^2$) clusters. Data as mean ± SD. n = 4–7, one-way ANOVA, *$P < 0.05$.

Taken together, these results show that when compared with controls, tauopathy flies subjected to sleep deprivation have a significantly higher accumulation of pTau deposition in the brain. In contrast, sleep induction reduced the accumulation of hyperphosphorylated pTau specifically in medium and large clusters and dampened pTau cluster accumulation in axons. When combined with the results observed in Fig 5, these results suggest altered protein processing after sleep modulation.

## Sleep modulation alters Tau aggregation and clearance

To uncover the mechanism underlying the reduction in pTau accumulation and the protective effects of sleep induction, we assessed the impact of sleep induction on Tau protein levels. To enable biochemistry analysis, we expressed Tau pan-neuronally using *elav-GAL4*. For the following experiments expressing Tau pan-neuronally, we first analyzed the sleep pattern and confirmed that these flies presented sleep disturbances as shown in Fig 1. After 4 d of sleep monitoring (Fig S4A), flies expressing Tau showed significant reduction in average

sleep (Fig S4B), increased sleep fragmentation (Fig S4C) and decreased sleep length (Fig S4D). We then subjected the flies to sleep modulation and analyzed their sleep profiles (Fig 7A). Quantification of sleep per 12 h showed significant sleep reduction after sleep deprivation for the normal control GFP group but showed no significant deprivation for Tau-expressing groups. This is likely because of the already significant sleep reduction and fragmentation because of Tau expression and the additional deprivation has reached the ceiling of the sleep disruption. In contrast, with sleep induction treatment, all groups showed significant sleep increase after gaboxadol feeding (Fig 7B), allowing detailed biochemical analysis of the potential neuroprotective effects of sleep induction. Western blot analysis of Tau expression levels (Fig 7C) showed no changes in protein levels in the deprived group but a significant decrease in total Tau (5A6) (Fig 7D) and hyperphosphorylated pTau, using S262 probe (Fig 7E) after sleep induction, consistent with behavioral data. The significant reductions in both total Tau and pTau levels through sleep induction suggest Tau protein clearance as a key mechanism underlying the beneficial effects of sleep increase.

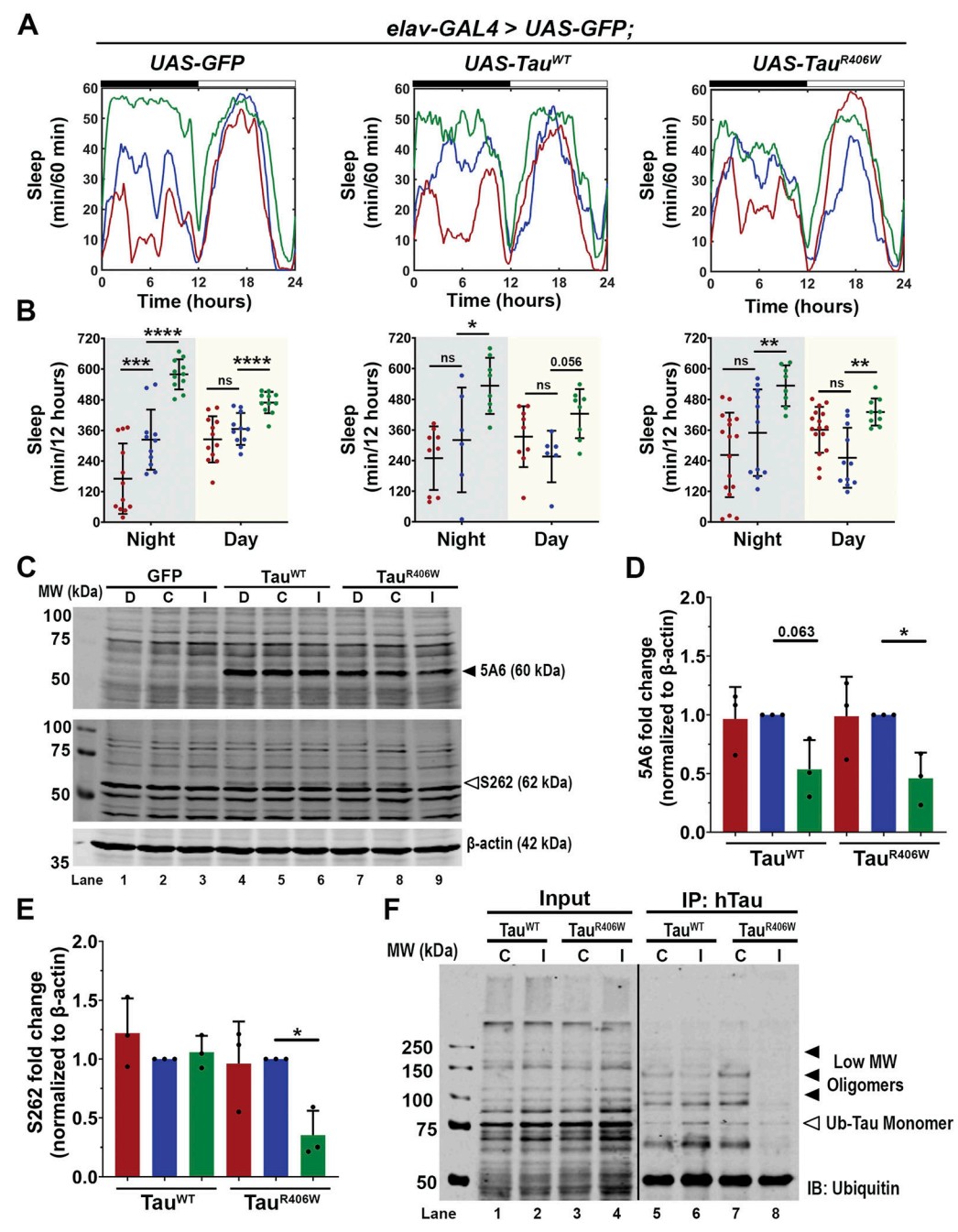

**Figure 7. Sleep induction promotes clearance of ubiquitinated Tau.**
**(A)** Sleep traces for 8 DAE flies expressing *UAS-GFP*, *UAS-hTau^WT* or *UAS-hTau^R406W* pan-neuronally using *elav-GAL4*. Sleep per 60 min traces showing control (blue), deprived (red), induced (green) groups. **(B)** Total sleep time per 12 h during night (gray box) and day (yellow box) for hours 132–156 is quantified. Data as mean ± SD. n = 6–17 flies, one-way ANOVA, *P < 0.05, **P < 0.01, ***P < 0.001, ****P < 0.0001. **(C)** Western blot probed for total Tau (5A6) (black arrowhead) and hyperphosphorylated Tau (S262) (white arrowhead). **(D)** Quantification of 5A6 fold change normalized to β-actin. **(E)** Quantification of S262 fold change normalized to β-actin. n = 3 biological replicates with 10 fly heads per group **(F)** Total hTau was immunoprecipitated from whole brain lysates of 8 DAE flies expressing either *Tau^WT* or *Tau^R406W* under control (C) or induction (I) and probed for ubiquitin. Blot shows low molecular weight oligomers (black arrowheads) that can be mono- or polyubiquitinated and ubiquitinated monomers (white arrowhead). Black line indicates a splice in the blot. n = 30 fly heads per group. Data as mean ± SD, one-way ANOVA. *P < 0.05.

Misfolded Tau proteins are usually degraded through multiple pathways such as the autophagy-lysosomal system and the ubiquitin-proteosome system (5). A common early step in these degradation pathways is the ubiquitination of Tau to mark the protein for degradation. Our previous study has shown the different oligomeric states of ubiquitinated Tau and identified the strong toxicity associated with low molecular weight Tau oligomers (29). We next examined the effects of sleep induction on the

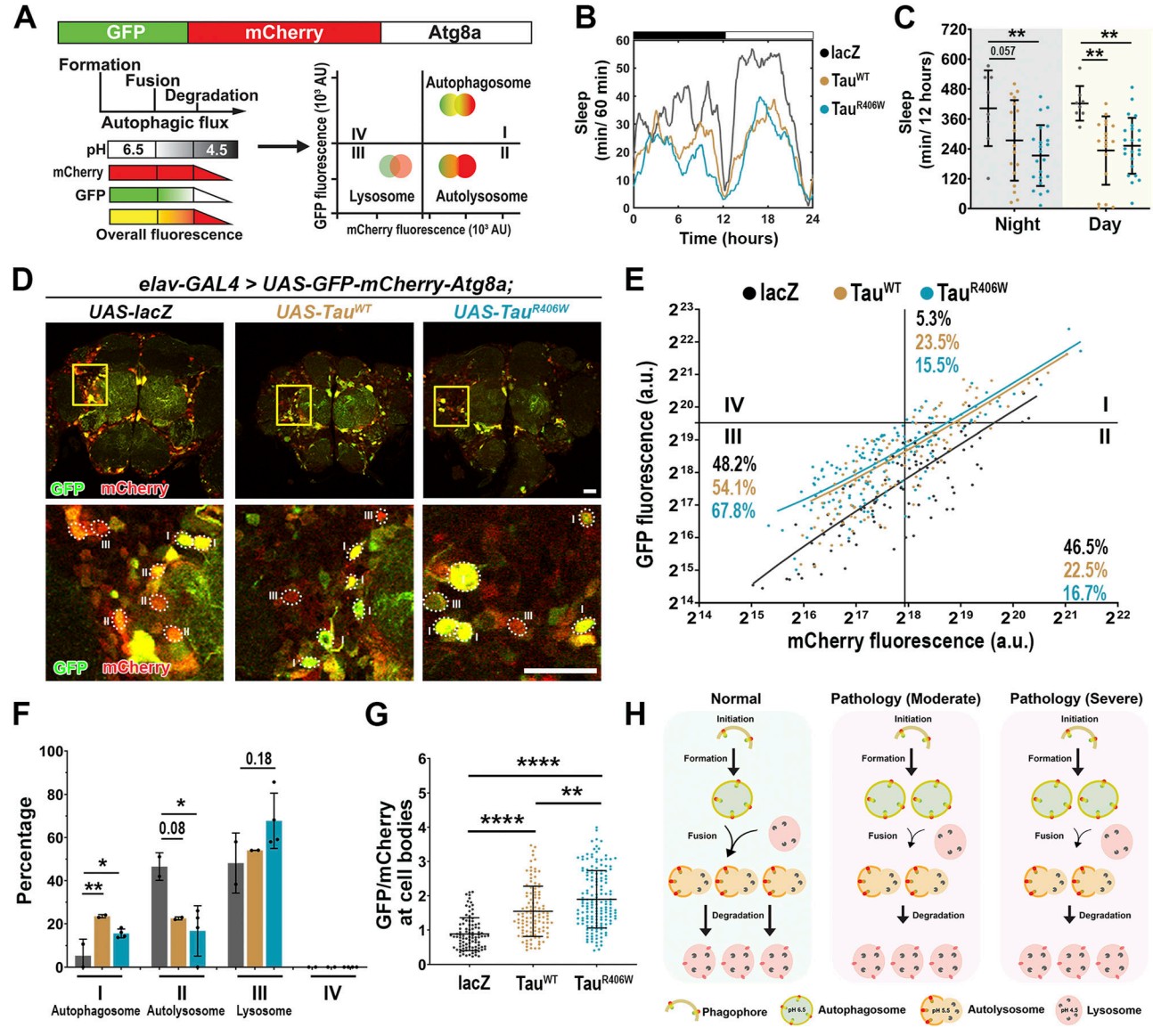

**Figure 8. Pan-neuronal Tau expression displays impaired autophagic flux.**
**(A)** Model of fluorescent reporter with a GFP (green fluorescent protein), mCherry (red fluorescent protein), and Atg8a (autophagy fusion protein). GFP expression is quenched under acidic pH. Fusion requires low pH therefore, cells undergoing autophagy will display more puncta marked with mCherry-Atg8a (autophagosomes and autolysosomes) than with GFP-Atg8 (autophagosomes only). **(B)** Sleep profiles for 8 DAE flies with pan-neuronal expression using *elav-GAL4* of *UAS-GFP-mCherry-Atg8a* fluorescent reporter with *UAS-lacZ* (gray), *UAS-Tau^{WT}* (gold), and *UAS-Tau^{R406W}* (light blue). **(C)** Total sleep time per 12 h during night (gray box) and day (yellow box) for hours 132–156 is quantified. Data as mean ± SD. n = 8–23, one-way ANOVA, **$P < 0.01$. **(D)** Confocal images of midbrain region showing GFP and mCherry fluorescence. Yellow box shows zoomed-in cell body region. Dotted circles highlight Atg8a positive neurons. Scale bar 30 $\mu m$. **(E)** GFP-mCherry-Atg8a puncta plotted by mean GFP intensity as a function of mean mCherry intensity. The plot was divided into four quadrants using gating thresholds of GFP = 750,000 and mCherry = 250,000. **(F)** Quantification of percentage of puncta in each quadrant; autophagosome stage (I), autolysosome (II), or lysosome (III). **(G)** Quantification of GFP/mCherry ratio. **(H)** Model showing that moderate and severe tauopathy impairs autophagic flux, causing increased autophagosome accumulation, decreased autophagosome-lysosome fusion, and decreased autolysosome degradation. Data as mean ± SD. n = 97–158 neurons from three to four fly brains, one-way ANOVA. *$P < 0.05$, **$P < 0.01$, ****$P < 0.0001$.

oligomerization of Tau species using biochemical approaches. Tau was immunoprecipitated by human Tau-specific antibody from lysates of Tau^{WT} or Tau^{R406W} expressing brains and probed from ubiquitin. Western blot analysis identified ubiquitinated Tau oligomeric species and found a reduction in low molecular weight toxic oligomers after sleep induction, when compared with control (Fig 7F). These results suggest that sleep induction alters Tau aggregation and promotes Tau clearance. These results are consistent

with previous reports showing that enhanced autophagy can suppress Tau accumulation (45).

## Tau expression impairs autophagic flux

A large body of literature has demonstrated the role of autophagy in neurodegeneration and highlighted the disturbance of autophagic flux in AD and related disorders (46, 47, 48, 49). To assess

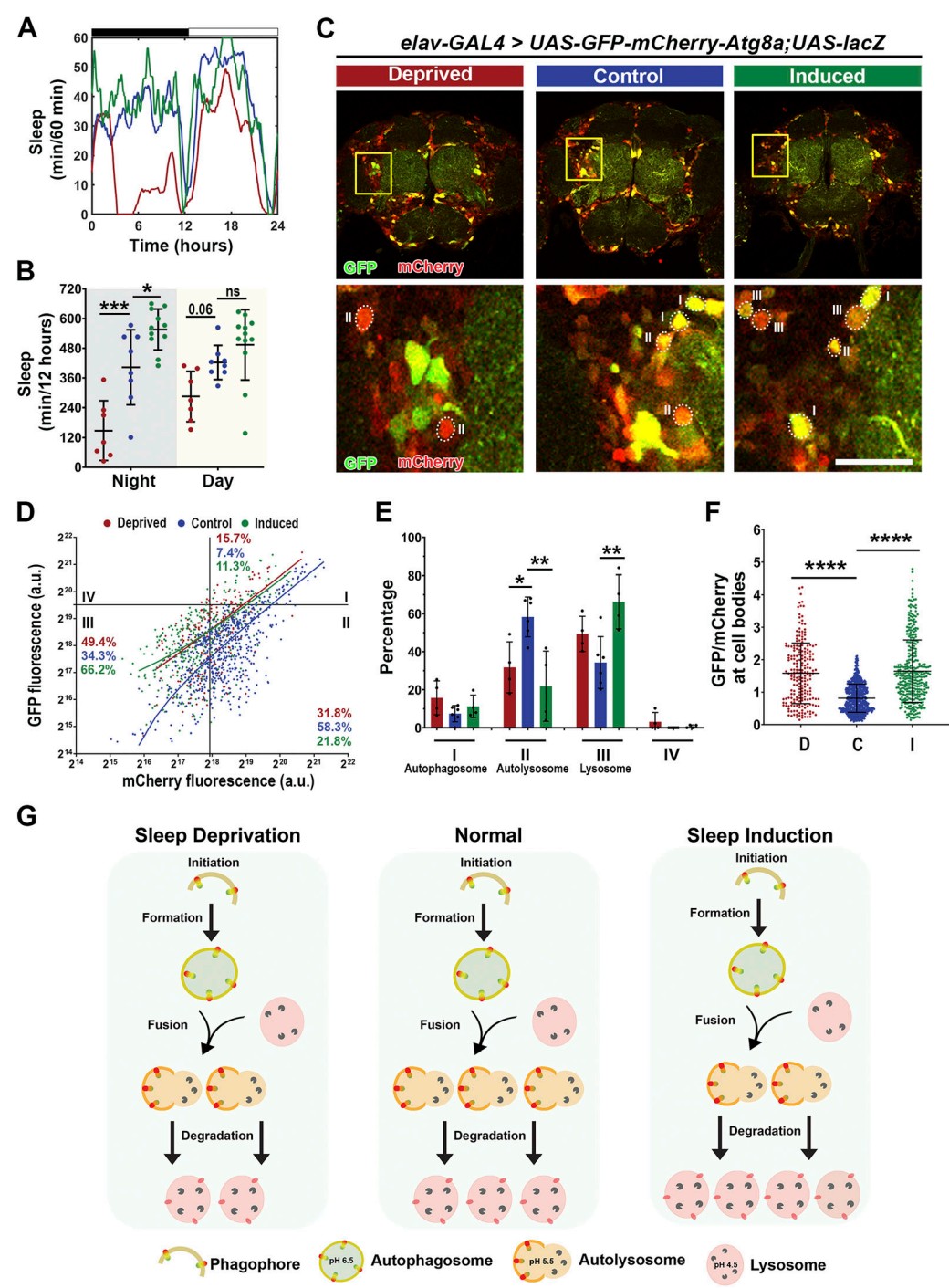

**Figure 9. Sleep modulation promotes autophagic flux in lacZ.**
**(A)** Sleep traces for 8 DAE flies with pan-neuronal expression using *elav-GAL4* of *UAS-GFP-mCherry-Atg8a* fluorescent reporter with *UAS-lacZ* under control (blue), deprived (red), or induced (green) conditions. **(B)** Total sleep time per 12 h during night (gray box) and day (yellow box) for hours 132–156 is quantified. Data as mean ± SD. n = 8–12, One-way ANOVA, *$P < 0.05$, ***$P < 0.001$. **(C)** Confocal images of midbrain region showing GFP and mCherry fluorescence. Yellow box shows zoomed-in cell body region. Dotted circles highlight Atg8a positive neurons. Scale bar 30 $\mu$m. **(D)** GFP-mCherry-Atg8a puncta plotted by mean GFP intensity as a function of mean mCherry intensity. The plot was divided into four quadrants using gating thresholds of GFP = 750,000 and mCherry = 250,000 for lacZ group. **(E)** Quantification of percentage of puncta in each quadrant for lacZ groups. **(F)** Quantification of GFP/mCherry ratio. **(G)** Model showing that under physiological conditions you have clearance of proteins through autophagy, whereas under sleep deprivation, there is decreased fusion of autophagosomes with lysosomes, and under sleep induction, there is increased flux, as shown by increased fusion and degradation. Data are presented as mean ± SD. n = 97–212 neurons from four to six fly brains, one-way ANOVA. *$P < 0.05$, **$P < 0.01$, ****$P < 0.0001$.

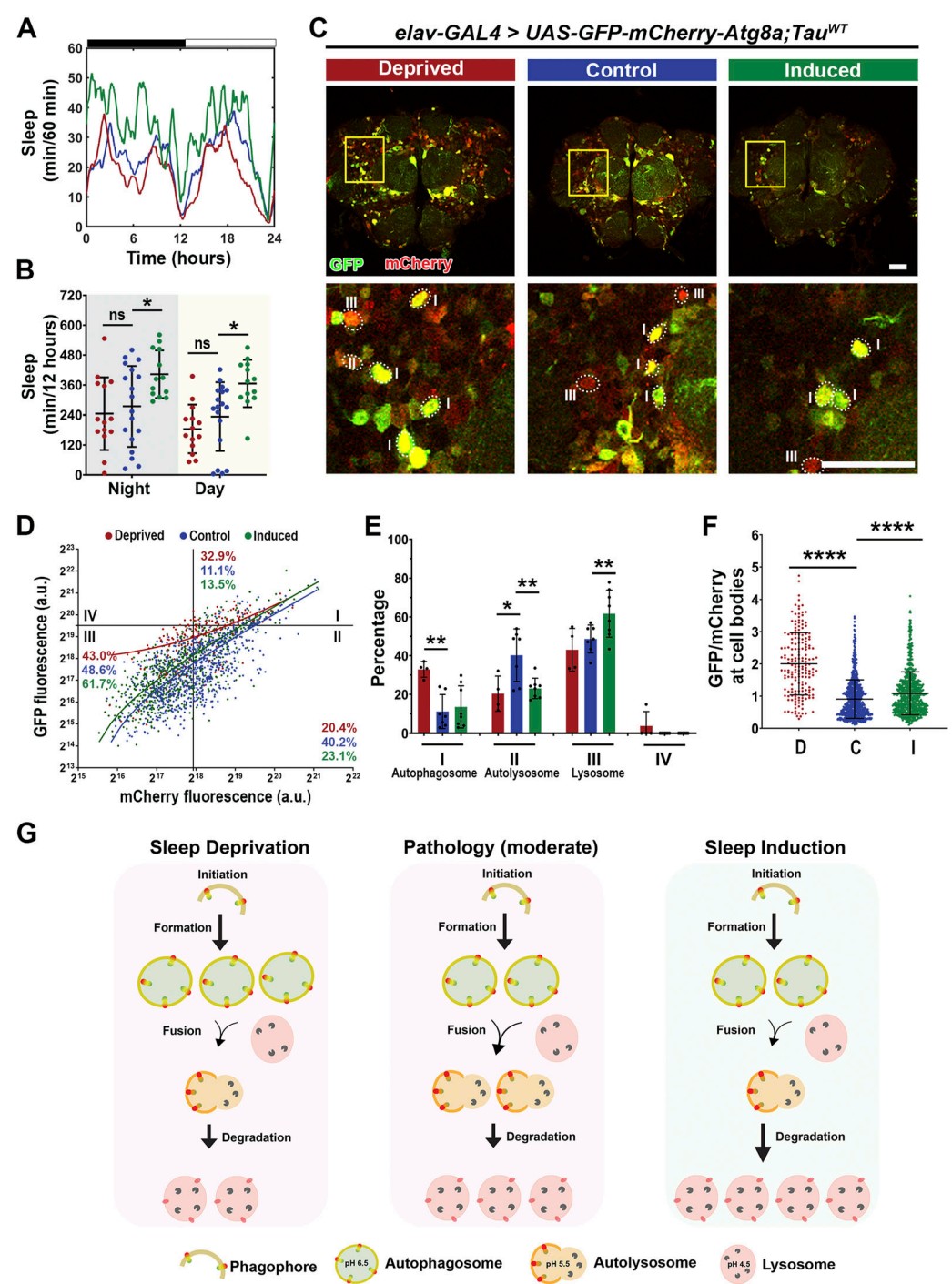

**Figure 10. Sleep induction modulates autophagic flux in Tau^WT expression flies.**
**(A)** Sleep traces for 8 DAE flies with pan-neuronal expression using *elav-GAL4* of *UAS-GFP-mCherry-Atg8a* fluorescent reporter with *UAS-Tau^WT* under control (blue), deprived (red), or induced (green) conditions. **(B)** Total sleep time per 12 h during night (gray box) and day (yellow box) for hours 132–156 is quantified. Data as mean ± SD. n = 13–18, one-way ANOVA, *$P < 0.05$. **(C)** Confocal images of midbrain region showing GFP and mCherry fluorescence. Yellow box shows zoomed-in cell body region. Dotted circles highlight Atg8a positive neurons. Scale bar 30 $\mu$m. **(D)** GFP-mCherry-Atg8a puncta plotted by mean GFP intensity as a function of mean mCherry intensity. **(E)** Quantification of percentage of puncta in each quadrant. **(F)** Quantification of GFP/mCherry ratio. **(G)** Model showing moderate pathology impairs autophagic flux, increasing autophagosome accumulation and decreasing formation of autolysosomes. Under sleep deprivation, we observe exacerbated impairment, as shown by increased autophagosome accumulation and decreased fusion, whereas after sleep induction, there is improved autophagic flux as shown by the increase in fusion and degradation. Data are presented as mean ± SD. n = 111–215 neurons from four to eight fly brains, one-way ANOVA, *$P < 0.05$, **$P < 0.01$.

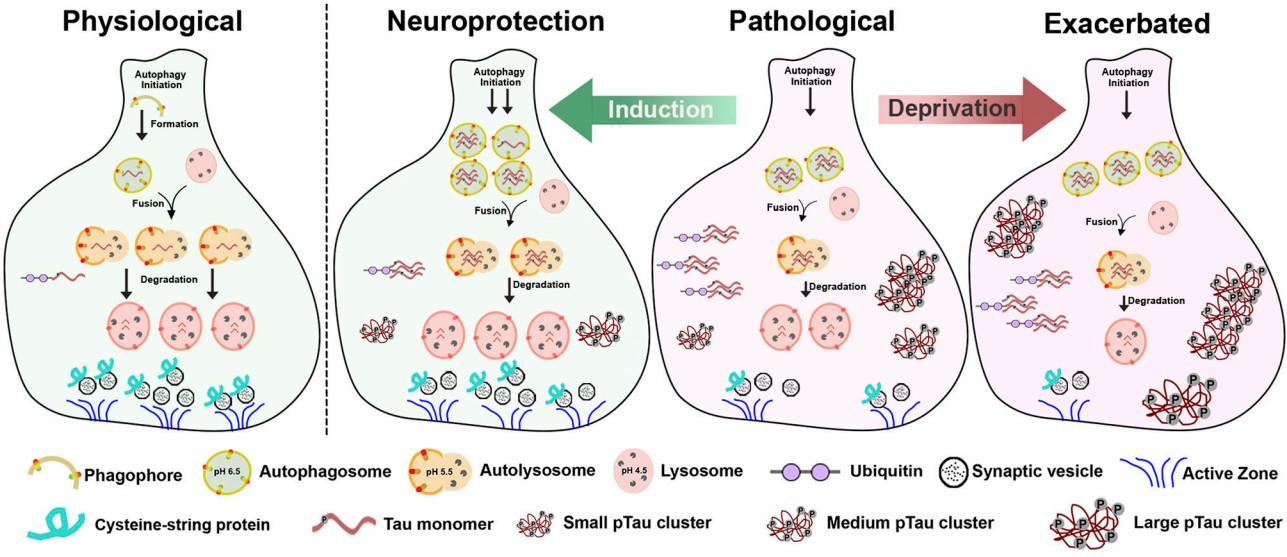

**Figure 11. Sleep modulation alters Tau aggregation, clearance, and synaptic degeneration in vivo in *Drosophila* models of tauopathy.**
Schematic model of synapse showing clearance of proteins through autophagy under physiological conditions (physiological). Pathological Tau expression results in impaired autophagic flux, accumulated hyperphosphorylated Tau clusters, and overall decreased synaptic integrity and function (pathological). Sleep deprivation leads to increased autophagosome accumulation and increased medium- and large-sized pTau clusters that result in exacerbated synaptic degeneration (exacerbated). In contrast, sleep induction confers neuroprotective effects as a result of increased autophagosome initiation, decreased medium and large-sized pTau clusters and improved synaptic integrity and function (neuroprotection).

how sleep induction impacts autophagic flux in tauopathy models, we incorporated an in vivo autophagy dual-color reporter expressing Atg8 tandem tagged with pH-sensitive GFP and mCherry at its amino-terminal (*UAS-GFP-mCherry-Atg8a*) (45, 50). Because GFP signal is quenched under the acidic environment, combination of GFP and mCherry signal intensity will mark the stages of autophagy flux, where autophagosomes will be high GFP and high mCherry, whereas fusion with the lysosome will be low GFP and high mCherry (Fig 8A).

We established autophagy reporter in tauopathy models by co-expressing *UAS-GFP-mCherry-Atg8a* with either *UAS-lacZ* (control) *UAS-Tau$^{WT}$* or *UAS-Tau$^{R406W}$* pan-neuronally using *elav-Gal4*. We first analyzed the baseline sleep profiles to assess whether reporter expression caused changes in sleep behavior (Fig 8B). Analysis of their activity behavior showed a significant reduction in nighttime and daytime sleep for Tau-expressing groups when compared with lacZ (Fig 8C). The activity profile is similar to the pan-neuronal tauopathy models (Fig 1A–D), suggesting that pan-neuronal Tau expression was the major driver of sleep disturbances, and the Atg8a reporter expression had minimum impact on sleep in tauopathy models.

Using confocal imaging and ratiometric-based quantification (44) we analyzed the fluorescence of Atg8-positive neuronal cell bodies in the fly midbrain region as described previously (44) (Fig 8D). The GFP and mCherry fluorescence intensity plotted is the combined fluorescence of all Atg8-positive organelles present per neuron. The results were plotted in XY for the GFP fluorescence (Y) as a function of mCherry fluorescence (X) (Fig 8E). Quadrants were divided based on intensity, where Quadrant I, high GFP-high mCherry marks autophagosome; Quadrant II, low GFP-high mCherry marks autolysosome; and Quadrant III, low GFP-low mCherry marks lysosomal degradation. The gating thresholds for

quadrants were set based on the control group using the following principles. (1) The design of tandem reporter of pH-sensitive GFP and pH-insensitive mCherry defines the stoichiometry of GFP/mCherry as ≤ 1. Therefore, the content of Quadrant IV (GFP/mCherry ≥ 1) is 0. Puncta with high GFP signal and no mCherry signal were likely the result of GFP-mCherry protein truncation and, therefore, were excluded from the analysis. (2) The distribution of Atg8-positive particles reflects the distribution of autophagy-lysosome pathway-related compartments in wild-type neurons (44, 51).

Compared with the lacZ control group, Tau$^{WT}$- and Tau$^{R406W}$-expressing neurons showed a significant increase in Quadrant I, autophagosome accumulation, and a significant reduction in Quadrant II, fusion of the autophagosome with the lysosome (Fig 8F). Moreover, the significant increase in GFP/mCherry ratio in Tau expression neurons suggests a higher pH level and dysfunction of the cellular organelles with acidic pH (Fig 8G). Together, these results reveal a shift in Atg8a puncta distribution in Tau pathology and further suggest a potential block in autophagy flux at the step of fusion of autophagosome and lysosome (Fig 8H). The finding of reduced fusion and a consequent increase in autophagosome accumulation and an increase in GFP/mCherry ratio would be consistent with Tau$^{WT}$ or Tau$^{R406W}$ expression-induced impairment of autophagic flux.

## Sleep induction confers neuroprotection through promoting Tau clearance

To examine the cellular impact of sleep modulation on autophagy flux, we first evaluated the control condition where lacZ and GFP-mCherry-Atg8 reporter were co-expressed. Sleep profiles confirmed

that sleep modulation was successful in Atg8 reporter expressing flies (Fig 9A), with a significant sleep decrease after sleep deprivation and a significant increase in sleep after induction (Fig 9B). Quantitative confocal imaging analysis of Atg8a puncta in the neuronal cell body region in the midbrain (Fig 9C), showed a significant shift in puncta distribution under sleep modulation (Fig 9D and E). Specifically, sleep deprivation caused a significant drop in Quadrant II, but no change in Quadrant III, whereas sleep induction caused a remarkable drop in Quadrant II and a concomitant increase in Quadrant III, suggesting a significant reduction in puncta in the fusion state, and a significant increase in puncta undergoing lysosomal degradation (Fig 9E). Moreover, there was a significant increase in GFP/mCherry ratio after sleep deprivation and induction (Fig 9F). An increase in ratio could be because of either increase in GFP because of increased autophagosome accumulation or a decrease in mCherry because of increased degradation. These results suggest that under normal (non-pathological) conditions, the effects of sleep deprivation on autophagy are minimal, whereas sleep induction appears to promote autophagic flux and may facilitate lysosomal degradation (Fig 9G).

Next, we analyzed how sleep modulation affects autophagic flux in tauopathy models. In Tau$^{WT}$ expression (moderate) model, sleep modulation was successful with sleep induction (Fig 10A and B). Imaging analysis of the Atg8a dual reporter expressed in the brain (Fig 10C) showed a significant increase in autophagosome puncta (Quadrant I) after sleep deprivation, accompanied by a decrease in Quadrant II. Moreover, after sleep induction a decrease in Quadrant II, with a concomitant increase in Quadrant III, was observed (Fig 10D and E). In addition, the GFP/mCherry ratio was increased after sleep modulation (Fig 10F). These results suggest that under moderate pathology sleep deprivation leads to increased autophagosome accumulation and decreased fusion with the lysosome, whereas under sleep induction, an increase in fusion and degradation occur (Fig 10G). In the Tau$^{R406W}$ expression (severe) model, sleep modulation was successful with sleep induction (Fig S5A and B), similar to that in the Tau$^{WT}$ expression (moderate) model (Fig 10A and B). However, analysis of autophagy flux using the Atg8a reporter revealed differences between the two pathological states (Fig S5C). Specifically, no significant changes were observed after sleep deprivation, whereas after sleep induction, we observed a decrease in Quadrant II (autolysosome) (Fig S5D and E) and an increase in GFP/mCherry ratio after sleep modulation (Fig S5F). These results suggest that under severe tauopathy pathology, sleep modulation had a limited effect on autophagic flux because of the system being severely degenerated. Although we observe minor changes after induction, there is high variability in the results (Fig S5G). Taken together, these results suggest that under moderate Tau pathology, sleep modulation promotes autophagic flux.

## Discussion

In this study, we established sleep modulation paradigms to examine the effect of altered sleep on the progression of Tau-induced neurodegeneration in *Drosophila* models of tauopathy. Using a multi-disciplinary approach of sleep behavior studies, genetic approaches, cellular and biochemical analysis, and physiological recordings, we discovered that sleep modulation alters Tau protein aggregation, clearance, and synaptic degeneration in vivo. We show that sleep deprivation enhanced Tau aggregational toxicity resulting in exacerbated synaptic degeneration. In contrast, sleep induction using gaboxadol led to reduced hyperphosphorylated Tau accumulation in neurons as a result of modulated autophagic flux and enhanced clearance of ubiquitinated Tau, suggesting altered protein processing and clearance that resulted in improved synaptic integrity and function. Specifically, pathological Tau expression impairs autophagic flux, accumulated hyperphosphorylated Tau clusters and overall decreased synaptic integrity and function. After sleep deprivation, there is increased autophagosome accumulation, decreased formation of autolysosomes and increased medium and large-sized pTau clusters that lead to exacerbated synaptic degeneration. In contrast, sleep induction exerts neuroprotective effects likely through increased autophagosome initiation, increased fusion and degradation, decreased medium- and large-sized pTau clusters, and improved synaptic integrity and function (Fig 11).

Sleep is essential for normal human function. It modulates synaptic plasticity; it is critical for memory consolidation, and it is essential for neurotoxic clearance of proteins from the brain (52). Lack of sleep can disturb circadian physiology and lead to impaired memory, elevated oxidative stress, and increased risk of disease (1, 15, 53). In AD, altered sleep occurs early and leads to an increased risk of developing cognitive impairment and AD progression.

Tau pathology is highly associated with synaptic loss and altered synaptic function and synapse loss is the main correlate with cognitive decline in AD (54, 55). In this study, we observed that sleep deprivation resulted in increased hyperphosphorylated Tau accumulation in axons. These results are consistent with increased evidence showing that sleep deprivation leads to neurotoxic protein accumulation such as Aβ and Tau (7, 33, 56, 57). This increased accumulation of Tau caused accelerated progression of synaptic degeneration as observed by the decreased thickness of the lamina neuropil and decrease presence of synaptic structural proteins at synapses. It is important to note that the effect of sleep modulation on protein homeostasis is not restricted to Tau protein, as indicated by the results of reporter membrane mCD8-GFP expression experiment. We observed an increase in mCD8-GFP accumulation after sleep deprivation and a decrease after sleep induction only in Tau-expressing neurons. The effect of sleep modulation on protein homeostasis in normal healthy neurons is small and insignificant, suggesting that the extent of our sleep modulation is within the normal neuronal homeostatic capacity. However, under pathological conditions, neuronal homeostatic buffering system is disrupted, and therefore, the effect of sleep modulation is amplified and highly significant. Taken together, these results further confirm the direct interplay between sleep disturbances and protein homeostasis especially under pathological conditions. Our data support increasing evidence suggesting a positive feedback loop where impairment of one mechanism, exacerbates the other, and together can accelerate AD progression.

Furthermore, sleep induction using gaboxadol, a GABA$_A$ receptor agonist, showed a substantial reduction in total Tau protein levels and hyperphosphorylated Tau. In addition, we

observed a reduction in Tau accumulation in synapses. This reduction resulted in improved thickness of the lamina neuropil, overall presence of active zone proteins, and improved synaptic function. It is important to note that the effect of gaboxadol has been shown to be mainly through increasing sleep, rather than GABA activation. Specifically, knockdown of GABA$_A$ receptors prevented long-term memory consolidation following gaboxadol administration (34). These results support the idea that sleep-enhancing treatments can delay Tau-induced neurodegeneration.

Autophagy plays a crucial role in protein homeostasis in the context of neurodegeneration (58). Our observation of Tau-induced impaired autophagic flux, shown by increased autophagosome accumulation and decreased fusion and lysosomal degradation, is consistent with prior research showing disturbed clearance of autophagic vacuoles in AD brain and APP mouse models (46) and increased autophagosome accumulation caused by MAPT overexpression in mice (47). Previous work has highlighted the connection between sleep and autophagy specifically showing that under unperturbed autophagy, sleep increases clearance of autophagosomes and/or decreases production of autophagosomes (20). Our data using the autophagy reporter provides evidence that under pathological conditions where autophagy is already disturbed, the effects of sleep modulation might differ from unperturbed conditions. Sleep induction using gaboxadol showed increased autophagosomes and led to decreased amounts of ubiquitinated low molecular weight Tau toxic oligomers. Together, these data suggest that sleep induction is promoting the formation of autophagosomes to push the flux forward and allow for more Tau degradation to occur.

With current treatments, neurodegeneration persists despite interventional treatment. Some clinical trials are now exploring the potential effects of sleep aids such as suvorexant to decrease the rate of A$\beta$ accumulation in the brain (59). Many of the studies have focused on sleep interventions with already established AD dementia, where pathology is already too advanced, making treatments ineffective against the progression of the disease (60). Our data suggest that sleep can affect proteostasis and the clearance of misfolded proteins, consequently the progression of disease from early stages. Future work is required to identify biomarkers that detect neuronal susceptibility to changes in protein homeostasis. Therefore, given the abundance of sleep-enhancing pharmaceuticals on the market, determining the neuroprotective effects of sleep-inducing agents on protein homeostasis before the onset of clinical symptoms is critical for early intervention.

# Materials and Methods

### Drosophila stocks and genetics

Flies were reared on cornmeal-molasses-yeast medium at 22°C, 65% humidity, with 12-h light/12-h dark cycles. The following strains were used in this study: UAS-Tau$^{WT}$, UAS-Tau$^{R406W}$ obtained from Dr. Mel Feany (29). *UAS-GFP-mCherry-Atg8a, GMR-GAL4*, and

*elav-GAL4* were obtained from Bloomington *Drosophila* Stock Center.

### Locomotor activity

Male flies were placed in single glass tubes as previously described (61). Locomotion was recorded using activity monitors (DAM2, Trikineticks Inc.). Monitors were placed in an environmental chamber maintained at 25°C and 70–80% relative humidity. In all experiments, flies were first entrained in 12-h light/12-h dark conditions for 2 d. After entrainment, locomotion was measured with 20 s binning for 4 d. Data from monitors were visualized and processed with custom-written Matlab (MathWorks Inc.) scripts.

### Sleep modulation

Mechanical deprivation was adapted from previously described Sleep Nullifying Apparatus methods (37). Flies were subjected to two episodes of 9 h of deprivation. For sleep induction flies were fed either vehicle or 0.1 mg/ml gaboxadol that was mixed with normal fly food (20).

### Drosophila CNS immunostaining, confocal imaging, and analysis

Fly brains were dissected in PBS (pH 7.4) as previously described (44). Samples were fixed in 4% formaldehyde for 10 min and washed in PBS containing 0.4% vol/vol Triton X-100 (PBTX). Samples were then incubated with primary antibodies diluted in 0.4% PBTX with 5% normal goat serum at 4°C overnight, followed by incubation with secondary antibodies diluted in 0.4% PBTX with 5% normal goat serum at 4°C overnight, and DAPI (1:300, D1306; Invitrogen) staining at room temperature for 10 min. The following primary antibodies were used: mouse anti-BRP (NC82,1:250; Developmental Studies Hybridoma Bank), mouse anti-CSP (6D6,1:250; Developmental Studies Hybridoma Bank), AT8 human PHF-Tau (MN1020, 1:250; Thermo Fisher Scientific). The following secondary antibodies were used: Alexa Fluor 555-conjugated anti-mouse (A21422, 1:300; Invitrogen) and Alexa Fluor 647 anti-mouse secondary antibody (A21235, 1:300; Invitrogen). The samples were mounted on glass slides with VECTASHIELD Antifade Mounting Medium (Vector Laboratories). Slides were imaged using an Olympus IX81 confocal microscope with a 40x or 60x oil immersion objective lens with a scan speed of 8.0 µs per pixel and spatial resolution of 1,024 × 1,024 pixels. Images were processed using FluoView 10-ASW (Olympus). Quantification of the number, area and intensity of fluorescently tagged proteins was carried out using ImageJ/Fiji (1.53q) using previously published methods (44). For quantification of tissue thickness of the lamina neuropil, three regions of interest (ROI) were averaged. For BRP intensity in the x-z plane, the lamina neuropil was chosen as the ROI. For intensity of the x-y plane in BRP and CSP staining, a 3 × 3 ROI was measured and divided by 9 to obtain average intensity per lamina cartridge.

For quantification of autophagic flux using tandem reporter previously published methods were followed (44, 51). The plot was divided into four quadrants using gating thresholds of GFP = 750,000 and mCherry = 250,000. The gating thresholds for quadrants were set based on the control group using the following principles.

(1) The design of tandem reporter of pH-sensitive GFP and pH-insensitive mCherry defines the stoichiometry of GFP/mCherry as ≤ 1. Therefore, the content of Quadrant IV (GFP/mCherry ≥ 1) is 0. (2) The distribution of Atg8-positive particles reflects the distribution of autophagy-lysosome pathway-related compartments in WT neurons (44, 51).

### ERG recordings and analysis

Previously described methods were followed (42, 62). Flights were anesthetized with $CO_2$ and immobilized on a glass slide. A recording electrode with 3 M NaCl was placed on the surface of the left eye, and another reference electrode was inserted into the thorax. After 5 min of dark adaptation, flies were given 1-s light stimulation (Digitimer), with 5 s of dark between each stimulus a total of 10 traces were recorded per fly and analyzed by pCLAMP 10 Electrophysiology Data Acquisition & Analysis Software (ver.10.5).

### Immunoprecipitation and Western blot analysis

Ten fly heads of each genotype were collected and flash-frozen before extraction. They were homogenized in radioimmunoprecipitation assay buffer (Sigma-Aldrich), protease inhibitor cocktail (Sigma-Aldrich) and a phosphatase inhibitor cocktail (Roche). For immuno-precipitation experiments, 30 8 DAE male fly heads were homogenized in lysis buffer, and incubated with Protein-G beads conjugated with 10 μg of antibody per sample overnight at 4°C; the bead pellets were collected and washed four times with lysis buffer before eluting out the beads fraction with 4x Lamelli buffer, heating the samples at 95°C and centrifuging for 5 min to collect the bead-eluted fraction. Lysates were probed with anti-human tau (5A6, 1:1,000; Developmental Studies Hybridoma Bank). For Western blotting, 10 μl of total protein lysate per sample (equivalent to 1 head) was loaded onto a polyacrylamide gel (PAGE) and resolved by SDS–PAGE and transferred onto nitrocellulose membrane. After blocking at room temperature for 1 h, the membrane was incubated with a primary antibody at 4°C overnight, followed by a secondary antibody for 1 h at room temperature. Imaging was performed on an Odyssey Infrared Imaging System (LI-COR Biosciences) and analyzed using Image Studio (v4.0). Primary antibody dilutions were used as follows: ubiquitin (3936, 1:1,000; Cell signaling), 5A6 anti-human tau (5A6, 1:250; Developmental Studies Hybridoma Bank), Phospho-Tau $Ser^{262}$ (44-750G, 1:1,000; Thermo Fisher Scientific), anti-β-actin (A1978, 1:5,000; Sigma-Aldrich). Secondary antibodies used were IRDYE800CW anti-mouse (610-145-002, 1:10,000; Rockland) and IRDYE700DX anti-rabbit (611-144-002, 1:10,000; Rockland).

### Statistical analyses

Biological sample size (n) and P-values are indicated in the corresponding figure legends. Either one-way ANOVA with Tukey post hoc test or Brown-Forsythe and Welch ANOVA with Dunnett's multiple comparison correction was applied to compare multiple groups. $P < 0.05$ was considered statistically significant. All statistical analyses were performed in GraphPad Prism software (version 10.0).

## Data Availability

All data are available in the main text or the supplementary materials.

## Supplementary Information

## Acknowledgements

We thank Zoraida Diaz-Perez for technical assistance. This work was supported in part by Florida Department of Health grant (FDOH 21A21) and National Institutes of Health grant (3R33AT010408-05S1) to RG Zhai, and the National Science Foundation grant (NSF# 2131037) to S Syed.

### Author Contributions

N Ortiz-Vega: investigation, visualization, methodology, and writing—original draft, review, and editing.
AG Lobato: investigation, methodology, and writing—review and editing.
T Canic: investigation, methodology, and writing—review and editing.
Y Zhu: investigation, methodology, and writing—review and editing.
S Lazopulo: investigation, methodology, and writing—review and editing.
S Syed: conceptualization, supervision, funding acquisition, investigation, methodology, and writing—review and editing.
RG Zhai: conceptualization, supervision, investigation, methodology, and writing—original draft, review, and editing.

### Conflict of Interest Statement

The authors declare that they have no conflict of interest.

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
