## [Reviewer comments · Life Science Alliance]

Life Science Alliance

Regulation of proteostasis by sleep through autophagy in *Drosophila* models of Alzheimer's Disease

Natalie Ortiz-Vega, Amanda Lobato, Tijana Canic, Yi Zhu, Stanislav Lazopulo, Sheyum Syed, and Rong Grace Zhai
DOI: <https://doi.org/10.26508/lsa.202402681>

Corresponding author(s): Rong Grace Zhai, University of Chicago

Review Timeline:	Submission Date:	2024-02-26
	Editorial Decision:	2024-04-12
	Revision Received:	2024-08-08
	Editorial Decision:	2024-08-27
	Revision Received:	2024-08-28
	Accepted:	2024-08-29

Transaction Report:

April 12, 2024

Re: Life Science Alliance manuscript #LSA-2024-02681-T

Dr. Rong Grace Zhai
University of Miami Health System
Molecular and Cellular Pharmacology
1600 NW 10th Ave
Miami, FL 33136

Dear Dr. Zhai,

Thank you for submitting your manuscript entitled "Regulation of proteostasis by sleep through autophagy in Drosophila models of Alzheimer's Disease" to Life Science Alliance. The manuscript was assessed by expert reviewers, whose comments are appended to this letter. We invite you to submit a revised manuscript addressing the Reviewer comments.

Thank you for this interesting contribution to Life Science Alliance. We are looking forward to receiving your revised manuscript.

Sincerely,

B. MANUSCRIPT ORGANIZATION AND FORMATTING:

Reviewer #1 (Comments to the Authors (Required)):

This paper investigates the relationship between sleep and proteosis dysregulation and investigated the effect of sleep deprivation or induction on Tau aggregation and toxicity. There is indeed a great interest in the link between sleep and neurodegenerative disease, not just at the molecular level (as this paper alludes) but also from a therapeutic perspective. Thus the subject matter of this paper is timely and will be of interest to others in the field. The data was well presented overall but I have some major concerns

Main concerns:

1. I failed to understand the rationale for expressing tau in photoreceptors to study the physiological impact of sleep modulation. The two are unrelated. Expression of tau in the different circuits will have circuit specific effects and findings from one circuit cannot be a pseudo indicator of the effect that may occur in the other circuit especially for the behaviour controlled by that circuit. Whilst I understand they wanted to look at the molecular events that underpin tau aggregation in a system that is not affected by sleep modulation, they did not make this clear in their text on page 3 and fig 1 and need to rewrite those paragraphs to make this clearer.
2. The data displayed in figure 2 is convincing about the different parameters that were employed to manipulate sleep; however the data showed in figure 3 is not convincing for several reasons - the images do not demonstrate reproducible and robust synaptic degeneration as BRP intensity is not an accurate measure of synaptic degeneration. Higher magnification images may show synaptic degeneration more clearly but the figures they present do not. Using BRP intensity is not the best way of measuring synaptic integrity as intensity can be altered by many factors that could be different from sample to sample and there is no evidence of an internal control that could have taken care of such differences. Though their ERG recordings do indeed show some sleep induced differences, I am not convinced that this is due to sleep related synaptic degeneration as I am not convinced that their data demonstrates robust synaptic degeneration.
3. Whilst there seems to be some effect of sleep deprivation and induction on the integrity of the photoreceptors as shown in Fig. 4, I am not entirely convinced that they can conclude that the phosphorylated tau levels, as evident by AT8 immunoreactivity intensity are changed - this is better depicted by a western blot of the eyes, which has been done by others using the GMR driver. Similarly, their filamentous tau cluster intensity is not necessarily an indicator of filamentous tau per se. Several publications have previously shown that expression of human tau, including these isoforms/mutants, leads to degeneration that is characterised by blebbing of axons due to axonal transport deficits in which multiple axonally transported cargo, including tau and mitochondria etc will accumulate. So these clusters are not necessarily comprised of filamentous tau and they cannot conclude that sleep deprivation or induction modulates filamentous tau formation in axons. It may be dampening tau-mediated degeneration and what they are viewing is a pseudo marker for this. If they want to demonstrate an increase in AT8 then a WB of the eyes in sleep deprived vs control vs induced situations is the best way to do this.
3. In figure 5, whilst their data shows a very slight reduction in total tau levels in R406W group following sleep induction, they cannot conclude that there is a reduction in pS262 levels as the correct manner to assess this is to normalise the pS262 levels to total tau. Otherwise a reduction in total tau will inevitably lead to a reduction in pS262 tau and that would not be surprising. Their pS262 bands certainly do not imply that there is a significant reduction in pS262 levels following sleep induction. The only convincing data on Fig 5 is the IP which does indeed appear to show that sleep induction reduces the amount of oligomeric tau species.
4. In Fig 8, though their dot plot is persuasive in showing that there are differences between the different quadrants (fig 7D), this does not equate to statistical significant changes when plotted - for most of the comparisons (Fig 7E); these patterns were not evident with wt tau and were actually the opposite for R406W tau - compare the change to quadrant III for control flies after sleep induction (Fig 7E - controls vs Fig 9E); would one not imagine that sleep induction would increase autophagic flux if it was the mechanism by which there was an improvement in the R406W phenotype? They imply that there should be more tau degradation through greater autophagosome formation during sleep induction but their data does not support these conclusions. Indeed their data shows reduced quadrant III (autophagosomal degradation) in R406W flies following sleep induction.

Whilst the authors seek to address a timely issue and there may be some data worthy of further exploration in their study, most of their data does not support their conclusions.

Reviewer #2 (Comments to the Authors (Required)):

In this manuscript, the authors examine the role of sleep levels on Tau induced toxicity in the adult brain. Using the fly model to perform both genetic and environmental perturbations, the authors identify that sleep status alters Tau phosphorylation, aggregation, clearance, and synaptic degeneration. In addition, the authors link sleep induction with improved protein processing and clearance suggesting proteostasis as a common mechanism between sleep modulation and tauopathy. Finally, the authors demonstrate that alterations in autophagy are present in both sleep modulation and Tau overexpression and sleep induction may ameliorate autophagy defects induced by neuronal Tau overexpression. The authors provide a mechanistic link helping explain the sleep defects observed in Alzheimer's disease patients and provide additional information on the link between sleep deprivation and neurodegenerative disease. The manuscript is well written, and the data was compelling. My only major issue is the interpretation of the autophagy data upon sleep induction in severe Tau[R406W] expressing animals (Figure 9) in which I will discuss further below.

Major critiques:

In Figure 3A, the authors claim that sleep modulation can alter synaptic structure in both control and Tau overexpressing brains. Although bruchpilot is a well studied marker of the synapse, the use of gross bruchpilot intensity as a readout of synaptic structure is not completely convincing. It would be more convincing if additional synaptic structure marker e.g synapsin or CSP was visualized to confirm the staining with Brp. This experiment could be completed in 1 month.

In Figure 5A, neuronal expression of Tau[WT] and Tau[R406W] do not exhibit significant sleep defects (blue lines in graph) under control sleep conditions. However, authors claim within the text that Tau[WT] and Tau [R406W] animals exhibit "significant sleep reduction and fragmentation due to Tau expression." Authors should explain the conflict between the data and text in *elav-GAL4> UAS-Tau* animals?

In figure 9, I am not convinced with the interpretation of the data, the authors observe an increase in autophagosomes in Tau[R406W] overexpressing brains after sleep induction. The authors claim that sleep induction leads to increased autophagy initiation. However, the data not show a concurrent increase in the autolysosomes in these neurons as one would expect if this is true. Plus, the increase in autophagosomes present class I neurons is sub-significant. It would be a more appropriate interpretation to state that sleep deprivation and induction does not have any significant effect on autophagy in the severe Tau[R406W] model due to severity of neurodegeneration induced by mutant Tau overexpression.

We would like to thank the reviewers for their thoughtful comments and insightful suggestions. We have followed the recommendations, carried out the experiment to fully address the reviewers' concerns, and significantly improved our description and interpretation to substantiate the significance of our findings.

We have made comprehensive and extensive revisions. In total, we added 4 new figures, substantially revised 7 figures. Here we include two tables to provide an overview of the revision: Table 1, a list of new and revised data figures included in the revision; Table 2, a summary of responses to the main points raised by reviewers. A detailed, point-by-point response to all of the issues raised by the reviewers was included following the tables. In the manuscript, major revisions in the text are marked by a blue font to highlight the new data, while minor corrections made throughout the rest of the manuscript are unmarked.

Table 1: Summary list of major changes to the figures.

	Changes/ Additions	Experiments	Responding to Reviewer
Figure 3	New panel A, B, E. Moved Panel B, E- G to Figure 4	Higher magnification images of lamina cross section. Quantified BRP intensity per lamina cartridge.	Reviewer 1, 2
Figure S3, supplement to Figure 3	New	Probed for cysteine-string protein (CSP) to confirm synaptic degeneration phenotype	Reviewers 1, 2
Figure 4	New – Original Panel B, E-G from Fig 3	Modified to accommodate new experimental results	Reviewers 1, 2
Figure 5	New	Examined membrane-targeted GFP accumulation in neurons	Reviewers 1, 2
Figure 6	Removed Panel E		Reviewer 1
Figure S4 supplement to Figure 7	New	Quantified sleep profiles for flies expressing Tau pan-neuronally.	Reviewer 2
Figure 8	Revised	Added GFP/mCherry ratio in Panel G, Revised model in Panel H	Reviewers 1,2
Figure 9	Revised D-G	Increased number of flies, added GFP/mCherry ratio, modified model	Reviewers 1, 2
Figure 10	Revised D-G	Increased number of flies, added GFP/mCherry ratio, modified model	Reviewers 1, 2
Figure S5 Supplement to Figure 10	Revised from original Figure 10	Increased number of flies, added GFP/mCherry ratio, modified model	Reviewers 1, 2
Figure 11	Revised	Modified to include new experimental results	

Table 2: Summary of results from revision experiments carried out in response to key comments raised by all reviewers.

Key Comments	Reviewers		Remedy
	1	2	
1. Clarify rationale of using photoreceptor expression of Tau for the study.	•		Expanded on rationale in Results Paragraph 1 and 2.
2. Data in Fig 3 was not convincing in demonstrating robust synaptic degeneration and BRP measurement by itself was not sufficient to conclude this. Include additional synaptic marker	•	•	Included higher magnification BRP image and additional quantification of lamina cartridges. (Fig. 3) Carried out CSP staining and quantification. (Fig. S3)
3. Fig 4 Filamentous Tau cluster intensity is not necessarily an indicator of filamentous tau per se. Use WB of eyes to confirm phosphorylated Tau level change.	•		Removed Figure 5 Panel E and updated results and conclusions. Examined how sleep affects general protein processing using endogenous membrane-targeted GFP as a readout. Fig 4 Included Fig R1-A showing WB Probed for AT8 using elav > Tau flies
4. Fig 5 Normalize pS262 levels to total Tau if concluding reduction of pS262 levels. pS262 band does not imply significant reduction.	•		Included Fig R1-B showing pS262/Total Tau ratio.
5. Fig 7-9 Dot plot (Panel D) is persuasive but does not equate statistical significance in Panel E . Data does not support conclusion of greater autophagosome formation after induction, due to reduced Quadrant III in R406W flies following sleep induction.	•	•	Increased number of flies to enhance power of analysis. Updated results in Fig 8-10, Fig S5.
6. Fig 5A Tau expression does not exhibit significant sleep defects. Explain conflict between data and text		•	Included sleep analysis of flies utilized for this experiment in Fig S5.

Point-by-point response to reviewers' comments:

Reviewer #1

This paper investigates the relationship between sleep and proteosis dysregulation and investigated the effect of sleep deprivation or induction on Tau aggregation and toxicity. There is indeed a great interest in the link between sleep and neurodegenerative disease, not just at the molecular level (as this paper alludes) but also from a therapeutic perspective. Thus the subject matter of this paper is timely and will be of interest to others in the field. The data was well presented overall but I have some major concerns.

We thank the reviewer for recognizing the significance of our study and their insightful comments. In this revised version, we have carried out additional experiments and comprehensively revised our results and conclusions.

Major comment #1:

I failed to understand the rationale for expressing tau in photoreceptors to study the physiological impact of sleep modulation. The two are unrelated. Expression of tau in the different circuits will have circuit specific effects and findings from one circuit cannot be a pseudo indicator of the effect that may occur in the other circuit especially for the behaviour controlled by that circuit. Whilst I understand the they wanted to look at the molecular events that underpin tau aggregation in a system that is not affected by sleep modulation, they did not make this clear in their text on page 3 and fig 1 and need to rewrite those paragraphs to make this clearer.

We recognized our deficiency in providing a clear description. The following rationale was incorporated into the **Figure 1 results description**:

This experiment was designed to examine the early effects of sleep modulation before the onset of severe neurodegeneration in Tauopathy models [1]. Collectively, combining pan-neuronal (*elav-GAL4*) or cell type specific expression, in this case, photoreceptor expression of Tau (*GMR-GAL4*) models with successful sleep modulation allows the mechanistic dissection of the impact of sleep modulation on neurodegeneration in *Drosophila* models of Tauopathy.

To focus on the effects of sleep modulation on neurodegeneration and minimize the influence of Tau expression on sleep, we employed an additional Tauopathy model, by expressing Tau in a subset of neurons with minimum impact on sleep behavior. To that end, we analyzed whether expressing Tau in the photoreceptors using *GMR-GAL4* caused sleep disturbances by monitoring their sleep behavior.

Together, this data suggests that expression of Tau in the photoreceptors causes minimum effect on sleep, making it a feasible model to examine the molecular impact of sleep modulation on Tau aggregation in a system with relatively normal sleep behavior. Moreover, in *Drosophila*, gaboxadol induces sleep through the GABA_A receptors, specifically, *Ligand-gated chloride channel homolog 3* (*Lcch3*) and the *GABA and glycine-like receptor* (*Grd*) [2]. Previous research has shown these receptors are present in interneurons postsynaptic to photoreceptors [33, 34]. Therefore, incorporating the expression of Tau in the photoreceptors, using *GMR-GAL4* as a driver, also allows to confirm that any synaptic effects observed when feeding gaboxadol are due to increased sleep and not caused by direct activation of GABA receptors, as GABA receptors are not expressed in photoreceptors.

Major comment #2:

The data displayed in figure 2 is convincing about the different parameters that were employed to manipulate sleep; however the data showed in figure 3 is not convincing for several reasons - the images do not demonstrate reproducible and robust synaptic degeneration as BRP intensity is not an accurate measure of synaptic degeneration. Higher magnification images may show synaptic degeneration more clearly but the figures they present do not. Using BRP intensity is not the best way of measuring synaptic integrity as intensity can be altered by many factors that could be different from sample to sample and there is no evidence of an internal control that could have taken care of such differences. Though their ERG recordings do indeed show some sleep induced differences, I am not convinced that this is due to sleep related synaptic degeneration as I am not convinced that their data demonstrates robust synaptic degeneration.

We thank the reviewer for bringing up this important point. We have included higher magnification images of the cross section of the lamina neuropil stained for BRP to show lamina cartridge organization. We have also included quantification of intensity per lamina cartridge (**Figure 3**). We quantified the average BRP intensity per lamina cartridge and found a Tau expression-induced, significant disruption of cartridge integrity and reduced BRP intensity per lamina cartridge. This reduction was exacerbated with sleep deprivation and significantly improved after sleep induction for both Tau^{WT} and Tau^{R406W} (**Figure 3E**).

We have also carried out analysis on a second endogenous synaptic marker cysteine-string protein (CSP), to further substantiate the observation of synaptic degeneration (**Figure S3**), as described below.

To further extend the analysis on the effects of sleep modulation on synaptic integrity, we analyzed the synaptic localization of endogenous cysteine-string protein (CSP), a synaptic vesicle-associated chaperone critical for neurotransmission [3] (**Figure S3A**). Quantification of CSP intensity per lamina cartridge revealed a significant reduction of CSP per lamina cartridge in both Tau^{WT} and Tau^{R406W} groups. Furthermore, after sleep deprivation a significant reduction of CSP was observed in Tau^{WT} group. Lastly, a significant increase was observed after sleep induction in Tau^{R406W} group (**Figure S3B**). Consistent with BRP levels, and the ERG recordings shown in **Figure 4**, these results suggest that sleep deprivation exacerbated overall synaptic loss and integrity and sleep induction provided significant improvement of synaptic structures.

Figure 3: Sleep modulation influences Tau-induced impaired synaptic integrity and morphology. (A) The three-dimensional structure of the *Drosophila* visual system showing the lamina, medulla and lobula. The x-z and x-y planes showing the photoreceptor terminals and lamina neurons are indicated. The organized lamina cartridges, including a higher magnification example and columnar photoreceptor neurons are shown in the x-z and x-y planes, respectively. Yellow dashed box shows 3x3 example of lamina cartridges used for quantification of x-y plane. **(B)** Lamina structures at 8 DAE containing endogenous mCD8-GFP (green), probed for BRP (magenta) and stained with DAPI (cyan). Yellow dashed line highlights lamina neuropil in x-z plane. Yellow dashed box highlights organized lamina cartridges in x-y plane. Scale bar 10 μ m. Cellular and functional analyses were performed at 160 hours. **(C)** Quantification of tissue thickness of lamina neuropil (yellow bars shown on panel B) **(D)** Quantification of BRP intensity in the x-z plane of the lamina neuropil normalized to GFP control (yellow dashed line). Brown-Forsythe and Welch ANOVA with Dunnett's multiple comparison correction. **(E)** Quantification of average BRP intensity per lamina cartridge in x-y plane (yellow dashed box). For dissections n = 5-11 for each group. Data are presented as mean \pm SD. One-way ANOVA, * $p < 0.05$, ** $p < 0.01$, **** $p < 0.0001$.

Figure S3: Sleep modulation alters Tau-induced synaptic degeneration. Flies were expressing either *UAS-CD8-GFP*, *UAS-hTau^{WT}*, or *UAS-hTau^{R406W}* in the photoreceptors using *GMR-GAL4* driver. **(A)** Lamina structures at 8 DAE containing endogenous mCD8-GFP (magenta), probed for cysteine-string protein (CSP) (cyan). Yellow dashed box highlights organized lamina cartridges in x-y plane. Scale bar 10µm. Cellular and functional analyses were performed at 160 hours. **(B)** Quantification of average CSP intensity per lamina cartridge in x-y plane (yellow dashed box). For dissections n = 4-5 for each group. Data are presented as mean ± SD. One-way ANOVA, *p<0.05.

Major comment #3:

Whilst there seems to be some effect of sleep deprivation and induction on the integrity of the photoreceptors as shown in Fig. 4, I am not entirely convinced that they can conclude that the phosphorylated tau levels, as evident by AT8 immunoreactivity intensity are changed - this is better depicted by a western blot of the eyes, which has been done by others using the GMR driver. Similarly, their filamentous tau cluster intensity is not necessarily an indicator of filamentous tau per se. Several publications have previously shown that expression of human tau, including these isoforms/mutants, leads to degeneration that is characterized by blebbing of axons due to axonal transport deficits in which multiple axonally transported cargo, including tau and mitochondria etc will accumulate. So these clusters are not necessarily comprised of filamentous tau and they cannot conclude that sleep deprivation or induction modulates filamentous tau formation in axons. It may be dampening tau-mediated degeneration and what they are viewing is a pseudo marker for this. If they want to demonstrate an increase in AT8 then a WB of the eyes in sleep deprived vs control vs induced situations is the best way to do this.

We agree with reviewer that a WB of the eyes would provide the most accurate confirmation of the imaging results observed. Unfortunately, the sleep behavior apparatus is limited to house 64 flies per experiment, which have to be divided into 9 experimental groups. Therefore, obtaining sufficient eye tissue that would provide the resolution required to observe significant protein level changes after sleep modulation is not feasible with a WB of the fly eyes.

To address the concern, we performed a WB of whole brain tissue from flies expressing either GFP, Tau^{WT} and Tau^{R406W} pan-neuronally using *e/av-GAL4* and probed for AT8. The results show the presence of the protein when compared to negative control GFP. Moreover, after quantification the protein fold change showed high variability between experimental replicates, therefore even though the data shows a trend of increased AT8 after sleep deprivation and decrease after sleep induction, these results were not statistically significant (**Figure R1**). The trend supports the imaging results observed in **Figure 6** as well as the results obtained previously in **Figure 7**, where we probed for hyperphosphorylated Tau (pS262) and total Tau (5A6).

To address the second point of filamentous Tau formation in axons, we have removed **panel E** from **Figure 6** to avoid over-conclusion about the formation of filamentous tau in axons and have modified the description to Tau clusters.

In addition, we performed an additional experiment to analyze how sleep modulation affected general protein processing using the reporter mCD8-GFP protein. In our tauopathy model, we co-expressed mCD8-GFP, a membrane-targeted GFP as a reporter to monitor the neuronal membrane integrity (**Figure 5A**). In the control group where only GFP was expressed, normal photoreceptor morphology is marked by GFP fluorescence, showing the neuronal membrane of photoreceptor neurons R1-6, which extend into lamina neuropil, and R7-R8 terminals which are observed in the medulla neuropil (**Figure 5B**). Importantly, the GFP intensity and localization pattern were largely unchanged by sleep modulation, suggesting a minimal effect of sleep modulation on protein homeostasis in normal healthy neurons. However, in Tau expressing flies, after sleep deprivation there was a significant increase of GFP in the lamina for the Tau^{R406W} group, and a significant reduction after sleep induction (**Figure 5C**). In addition, quantification of the intensity at the R7/R8 terminals in the medulla neuropil, shows significant increase after sleep deprivation in Tau^{WT} group and significant reduction after sleep induction in Tau^{R406W} group (**Figure 5D**). Taken together these results show that sleep modulation had minimal effects on reporter GFP homeostasis in normal conditions, while significantly alters protein homeostasis under pathological Tau expressing conditions. The increase of reporter GFP accumulation after sleep deprivation could indicate a block in protein processing and toxic protein buildup in Tauopathy conditions; while the decrease of reporter GFP after sleep induction could suggest enhanced clearance of misfolded GFP proteins and reduction of toxic protein burden.

Figure 5: Sleep modulation alters membrane-targeted GFP accumulation in neurons when co-expressed with Tau. (A) *Drosophila* optic lobe was scanned. White boxes represent higher magnification of lamina (left) and medulla (right). Scale bar 10 μ m. **(B)** Pink dashed line shows lamina neuropil showing photoreceptor synaptic terminals R1-6, while blue dashed line shows synaptic terminals of photoreceptors R7/R8. **(C)** Intensity of GFP in lamina neuropil normalized to GFP control. One-way ANOVA. **(D)** Quantification of intensity of GFP in medulla neuropil. Data as mean \pm SD. $n = 5-11$, Brown-Forsythe and Welch ANOVA with Dunnett's multiple comparison correction, * $p < 0.05$, ** $p < 0.01$.

Figure 6: Sleep modulation alters hyperphosphorylated Tau accumulation in neurons. (A) *Drosophila* optic lobe was stained with hyperphosphorylated Tau antibody AT8 (Ser²⁰², Thr²⁰⁵) (heatmap 0-4095). Yellow box represents zoomed in region of interest used for quantification. Scale bar 30μm. Yellow arrowheads represent AT8 clusters quantified in panels C and D. **(B)** Quantification of total AT8 intensity of optic lobe. **(C)** Quantification of intensity of AT8 clusters. **(D)** Quantification of number of AT8 clusters divided in small (0.4<5 μm²), medium (5-10 μm²), and large (>10 μm²) clusters. Data as mean ± SD. n = 4-7, One-way ANOVA, *p<0.05.

Major comment #4:

In figure 5, whilst their data shows a very slight reduction in total tau levels in R406W group following sleep induction, they cannot conclude that there is a reduction in pS262 levels as the correct manner to assess this is to normalise the pS262 levels to total tau. Otherwise a reduction in total tau will inevitably lead to a reduction in pS262 tau and that would not be surprising. Their pS262 bands certainly do not imply that there is a significant reduction in pS262 levels following sleep induction. The only convincing data on Fig 5 is the IP which does indeed appear to show that sleep induction reduces the amount of oligomeric tau species.

We recognized our deficiency in clarity. To clarify, our data suggest a reduction of Tau protein, not a reduction of Tau phosphorylation. We have included **Figure R1B** to show the pTau/Total Tau ratio, which shows no significant change with sleep modulation, suggesting that sleep modulation does not specifically affect Tau phosphorylation. Therefore, we conclude that the changes observed are due to a loss of overall Tau protein, not phosphorylation per se.

Regarding the second point that the pS262 band does not show reduction following sleep induction, there might be some confusion as to which band was quantified due to the near placement of a strong band below it. We have included a yellow dashed box in the figure for clarification on the location of the band quantified. The WB shown as example, shows significant reduction of the sleep induced Tau^{R406W} group.

Figure 7: Sleep induction promotes clearance of ubiquitinated Tau. (A) Sleep traces for 8 DAE flies expressing *UAS-GFP*, *UAS-hTau^{WT}* or *UAS-hTau^{R406W}* pan-neuronally using *elav-GAL4*. Sleep per 60 minutes traces showing control (blue), deprived (red), induced (green) groups. (B) Total sleep time per 12 hours during night (gray box) and day (yellow box) for Hours 132-156 is quantified. Data as mean \pm SD. $n = 6-17$ flies, One-way ANOVA, * $p < 0.05$, ** $p < 0.01$, *** $p < 0.001$, **** $p < 0.0001$. (C) Western blot probed for total tau (5A6) (black

arrowhead) and hyperphosphorylated tau (S262) (white arrowhead). **(D)** Quantification of 5A6 fold change normalized to β -actin. **(E)** Quantification of S262 fold change normalized to β -actin. $n = 3$ biological replicates with 10 fly heads per group **(F)** Total hTau was immunoprecipitated from whole brain lysates of 8 DAE flies expressing either Tau^{WT} or Tau^{R406W} under control (C) or induction (I) and probed for ubiquitin. Blot shows low molecular weight (LMW) oligomers (black arrowheads) that can be mono- or polyubiquitinated and ubiquitinated monomers (white arrowhead). $n = 30$ fly heads per group. Data as mean \pm SD, One-way ANOVA, $*p < 0.05$.

Figure R1: (A) Western blot probed for AT8 **(B)** Quantification of AT8 fold change normalized to β -actin. **(C)** Quantification of S262 fold change normalized to β -actin. $n = 3$ biological replicates with 10 fly heads per group. Data as mean \pm SD, One-way ANOVA.

Major comment #5:

In Fig 8, though their dot plot is persuasive in showing that there are differences between the different quadrants (fig 7D), this does not equate to statistically significant changes when plotted - for most of the comparisons (Fig 7E); these patterns were not evident with wt tau and were actually the opposite for R406W tau - compare the change to quadrant III for control flies after sleep induction (Fig 7E - controls vs Fig 9E); would one not imagine that sleep induction would increase autophagic flux if it was the mechanism by which there was an improvement in the R406W phenotype? They imply that there should be more tau degradation through greater autophagosome formation during sleep induction but their data does not support these conclusions. Indeed their data shows reduced quadrant III (autophagosomal degradation) in R406W flies following sleep induction.

Whilst the authors seek to address a timely issue and there may be some data worthy of further exploration in their study, most of their data does not support their conclusions.

We thank the reviewer for pointing this out. To address this, we performed additional experiments to increase the number of flies and updated the original figures 6-9, now updated to **Figures 8-10 and Figure S5**.

We also want to clarify that the results presented are of a snapshot in time of dissection and that the GFP and mCherry fluorescence intensity plotted is the pooled fluorescence of all Atg8 positive organelles present in each neuron, i.e. each data point represents the summation of all Atg8 containing organelle within one neuron at the time of dissection. The scattered plot shows the distribution of the neuronal population that expresses Tau under sleep modulation.

Moreover, we have included an additional measurement of GFP/mCherry ratio within a single cell, which serves as a proxy of the autophagic flux in the cell: a higher GFP/mCherry ratio is the result of either higher GFP signal due to a dysfunction of the cellular organelles with acidic pH, or lower mCherry signal, due to an increase in degradation, or a combination of both.

It is important to note that autophagic flux is a dynamic process where the rate of initiation vs degradation differs from cell to cell and time to time. To characterize such dynamic cellular process, we employed multiple independent approaches, (1) biochemical analysis of endogenous protein processing (WB and IP), (2) immunostaining of protein localization and expression, and (3) fluorescent reporter-based analyses. Based on the combined results of these approaches, we formulated our conclusions and presented our working models (Figure 11).

The revised results on autophagic flux are the following:

Figure 8: Pan-neuronal Tau expression displays impaired autophagic flux.

Compared to the lacZ control group, Tau^{WT} and Tau^{R406W} expressing neurons showed a significant increase in Quadrant I, autophagosome accumulation, and a significant reduction in Quadrant II, fusion of the autophagosome with the lysosome (**Figure 8F**). Moreover, the significant increase in GFP/mCherry ratio in Tau expression neurons suggests a higher pH level likely due to a dysfunction of the cellular organelles with acidic pH (**Figure 8G**). Together, these results reveal a shift in Atg8a puncta distribution in Tau pathology, and further suggest a potential block in autophagy flux at the step of fusion of autophagosome and lysosome (**Figure 8H**). The finding of reduced fusion and a consequent increase in autophagosome accumulation and an increase in GFP/mCherry ratio, would be consistent with Tau^{WT} or Tau^{R406W} expression-induced impairment of autophagic flux.

Figure 9: Sleep modulation promotes autophagic flux in lacZ.

Quantitative confocal imaging analysis of Atg8a puncta in the neuronal cell body region in the midbrain (**Figure 9C**), showed significant shift in puncta distribution under sleep modulation (**Figure**

9D-9E). Specifically, sleep deprivation caused a significant drop in Quadrant II, but no change in Quadrant III, while sleep induction caused a remarkable drop in Quadrant II and a concomitant increase in Quadrant III, suggesting a significant reduction in puncta in the fusion state, and a significant increase in puncta undergoing lysosomal degradation, (**Figure 9E**). Moreover, there was significant increase in GFP/mCherry ratio after sleep deprivation and induction (**Figure 9F**). An increase in ratio could be due to either increase in GFP due to increased autophagosome accumulation or a decrease in mCherry due to increased degradation. These results suggest that under normal (non-pathological) conditions, the effects of sleep deprivation on autophagy are minimal, while sleep induction appears to promote autophagic flux and may facilitate lysosomal degradation (**Figure 9G**).

Figure 10: Sleep induction modulates autophagic flux in TauWT expression flies.

Imaging analysis of the Atg8a dual reporter expressed in the brain (**Figure 10C**) showed a significant increase in autophagosome puncta (Quadrant I) after sleep deprivation, accompanied by a decrease in Quadrant II. Moreover, after sleep induction a decrease in Quadrant II, with a concomitant increase in Quadrant III, was observed (**Figure 10D-10E**). In addition, the GFP/mCherry ratio was increased after sleep modulation (**Figure 10F**). These results suggest that under moderate pathology sleep deprivation leads to increased autophagosome accumulation and decreased fusion with the lysosome, while under sleep induction an increase in fusion and degradation occur (**Figure 10G**).

Figure S5: Sleep induction modulates autophagic flux in TauR406W expression flies.

However, analysis of autophagy flux using the Atg8a reporter revealed differences between the two pathological states (**Figure S5C**). Specifically, no significant changes were observed after sleep deprivation, while after sleep induction, we observed a decrease in Quadrant II (autolysosome) (**Figure S5D-S5E**) as well as an increase in GFP/mCherry ratio after sleep modulation (**Figure S5F**). These results suggest that under severe Tauopathy pathology, sleep modulation had a limited effect on autophagic flux, due to the system being severely degenerated. Although we observe minor changes after induction, there is high variability in the results (**Figure S5G**). Taken together, these results suggest that under moderate Tau pathology, sleep modulation promotes autophagic flux.

Figure 8: Pan-neuronal Tau expression displays impaired autophagic flux. (A) Model of fluorescent reporter with a GFP (green fluorescent protein), mCherry (red fluorescent protein), and Atg8a (autophagy fusion protein). GFP expression is quenched under acidic pH. Fusion requires low pH therefore, cells undergoing autophagy will display more puncta marked with mCherry-Atg8a (autophagosomes and autolysosomes) than with GFP-Atg8 (autophagosomes only). (B) Sleep profiles for 8 DAE flies with pan-neuronal expression using *elav-GAL4* of *UAS-GFP-mCherry-Atg8a* fluorescent reporter with *UAS-lacZ* (gray), *UAS-Tau^{WT}* (gold) and *UAS-Tau^{R406W}* (light blue). (C) Total sleep time per 12 hours during night (gray box) and day (yellow box) for Hours 132-156 is quantified. Data as mean \pm SD. $n = 8-23$, One-way ANOVA, $**p < 0.01$. (D) Confocal images of midbrain region showing GFP and mCherry fluorescence. Yellow box shows zoomed in cell body region. Dotted circles highlight Atg8a positive neurons. Scale bar 30 μ m. (E) GFP-mCherry-Atg8a puncta plotted by mean GFP intensity as a function of mean mCherry intensity. The plot was divided into four quadrants using gating thresholds of GFP = 750000 and mCherry = 250000. (F) Quantification of percentage of puncta in each quadrant; autophagosome stage (I), autolysosome (II) or lysosome (III). (G) Quantification of GFP/mCherry ratio. (H) Model showing that moderate and severe tauopathy impairs autophagic flux, causing increased autophagosome accumulation, decreased autophagosome-lysosome fusion and decreased autolysosomes degradation. Data as mean \pm SD. $n = 97-158$ neurons from 3-4 fly brains, One-way ANOVA, $*p < 0.05$, $**p < 0.01$, $****p < 0.0001$.

Figure 9: Sleep modulation promotes autophagic flux in lacZ. (A) Sleep traces for 8 DAE flies with pan-neuronal expression using *elav-GAL4* of *UAS-GFP-mCherry-Atg8a* fluorescent reporter with *UAS-lacZ* under control (blue), deprived (red) or induced (green) conditions. (B) Total sleep time per 12 hours during night (gray box) and day (yellow box) for Hours 132-156 is quantified. Data as mean \pm SD. n = 8-12, One-way ANOVA, * $p < 0.05$, *** $p < 0.001$. (C) Confocal images of midbrain region showing GFP and mCherry fluorescence. Yellow box shows zoomed in cell body region. Dotted circles highlight Atg8a positive neurons. Scale bar 30 μ m. (D) GFP-mCherry-Atg8a puncta plotted by mean GFP intensity as a function of mean mCherry intensity. The plot was divided into four quadrants using gating thresholds of GFP = 750000 and mCherry = 250000 for lacZ group. (E) Quantification of percentage of puncta in each quadrant for lacZ groups. (F) Quantification of GFP/mCherry ratio. (G) Model showing that under physiological conditions you have clearance of proteins through autophagy, while under sleep deprivation there is decreased fusion of autophagosomes with lysosomes, and under sleep induction there is increased flux, as shown by increased fusion and degradation. Data are presented as mean \pm SD. n = 97-212 neurons from 4-6 fly brains, One-way ANOVA, * $p < 0.05$, ** $p < 0.01$, **** $p < 0.0001$.

Figure 10: Sleep induction modulates autophagic flux in Tau^{WT} expression flies. (A) Sleep traces for 8 DAE flies with pan-neuronal expression using *elav-GAL4* of *UAS-GFP-mCherry-Atg8a* fluorescent reporter with *UAS-Tau^{WT}* under control (blue), deprived (red) or induced (green) conditions. (B) Total sleep time per 12 hours during night (gray box) and day (yellow box) for Hours 132-156 is quantified. Data as mean \pm SD. n = 13-18, One-way ANOVA, * $p < 0.05$. (C) Confocal images of midbrain region showing GFP and mCherry fluorescence. Yellow box shows zoomed in cell body region. Dotted circles highlight Atg8a positive neurons. Scale bar 30 μ m. (D) GFP-mCherry-Atg8a puncta plotted by mean GFP intensity as a function of mean mCherry intensity. (E) Quantification of percentage of puncta in each quadrant. (F) Quantification of GFP/mCherry ratio. (G) Model showing moderate pathology impairs autophagic flux, increasing autophagosome accumulation and decreasing formation of autolysosomes. Under sleep deprivation we observe exacerbated impairment, as shown by increased autophagosome accumulation and decreased fusion, while after sleep induction there is improved autophagic flux as shown by increase in fusion and degradation. Data are presented as mean \pm SD. n = 111-215 neurons from 4-8 fly brains, One-way ANOVA, * $p < 0.05$, ** $p < 0.01$.

C *elav-GAL4 > UAS-GFP-mCherry-Atg8a;UAS-Tau^{R406W}*

Figure S5: Sleep induction modulates autophagic flux in Tau^{R406W} expression flies. (A) Sleep traces for 8 DAE flies with pan-neuronal expression using *elav-GAL4* of *UAS-mCherry-GFP-Atg8a* fluorescent reporter with *UAS-Tau^{R406W}* under control (blue), deprived (red) or induced (green) conditions. (B) Total sleep time per 12 hours during night (gray box) and day (yellow box) for Hours 132-156 is quantified. Data as mean \pm SD. n = 10-23, One-way ANOVA, *** $p < 0.001$. (C) Confocal images of midbrain region showing GFP and mCherry fluorescence. Yellow box shows zoomed in cell body region. Dotted circles highlight Atg8a puncta. Scale bar 30 μ m. (D) mCherry-GFP-Atg8a puncta plotted by mean GFP intensity as a function of mean mCherry intensity. (E) Quantification of percentage of puncta in each quadrant. (F) Model showing that under severe pathology you have impaired autophagic flux, shown as increased autophagosome accumulation and decreased fusion of autolysosomes and with improvement in autophagic flux after sleep induction. Data are presented as mean + SD. n = 96-158 neurons from 3-7 fly brains, One-way ANOVA, * $p < 0.05$, *** $p < 0.0001$.

Reviewer #2

In this manuscript, the authors examine the role of sleep levels on Tau induced toxicity in the adult brain. Using the fly model to perform both genetic and environmental perturbations, the authors identify that sleep status alters Tau phosphorylation, aggregation, clearance, and synaptic degeneration. In addition, the authors link sleep induction with improved protein processing and clearance suggesting proteostasis as a common mechanism between sleep modulation and tauopathy. Finally, the authors demonstrate that alterations in autophagy are present in both sleep modulation and Tau overexpression and sleep induction may ameliorate autophagy defects induced by neuronal Tau overexpression. The authors provide a mechanistic link helping explain the sleep defects observed in Alzheimer's disease patients and provide additional information on the link between sleep deprivation and neurodegenerative disease. The manuscript is well written, and the data was compelling. My only major issue is the interpretation of the autophagy data upon sleep induction in severe Tau[R406W] expressing animals (Figure 9) in which I will discuss further below.

We thank the reviewer for recognizing the significance of our study and providing insightful suggestions. In this revised version, we have carried out additional experiments and comprehensively revised our results and conclusions.

Major comment #1:

In Figure 3A, the authors claim that sleep modulation can alter synaptic structure in both control and Tau overexpressing brains. Although bruchpilot is a well studied marker of the synapse, the use of gross bruchpilot intensity as a readout of synaptic structure is not completely convincing. It would be more convincing if additional synaptic structure marker e.g synapsin or CSP was visualized to confirm the staining with Brp. This experiment could be completed in 1 month.

We thank the reviewer for bringing up this important point. We have included higher magnification images of the cross section of the lamina neuropil stained for BRP to show lamina cartridge organization. We have also included quantification of intensity per lamina cartridge (**Figure 3**). We quantified the average BRP intensity per lamina cartridge and found a Tau expression-induced, significant disruption of cartridge integrity and reduced BRP intensity per lamina cartridge. This reduction was exacerbated with sleep deprivation and significantly improved after sleep induction for both Tau^{WT} and Tau^{R406W} (**Figure 3E**)

We have followed reviewer's suggestion of examining additional synaptic markers and carried out analysis of cysteine-string protein (CSP). The results are included in **Figure S3** and described as follows.

To further extend the analysis on the effects of sleep modulation on synaptic integrity, we analyzed the synaptic localization of endogenous cysteine-string protein (CSP), a synaptic vesicle-associated chaperone critical for neurotransmission [3] (**Figure S3A**). Quantification of CSP intensity per lamina cartridge revealed a significant reduction of CSP per lamina cartridge in both Tau^{WT} and Tau^{R406W} groups. Furthermore, after sleep deprivation a significant reduction of CSP was observed in Tau^{WT} group. Lastly, a significant increase was observed after sleep induction in Tau^{R406W} group (**Figure S3B**). Consistent with BRP levels, and the ERG recordings shown in **Figure 4**, these results suggest that sleep deprivation exacerbated overall synaptic loss and integrity and sleep induction provided significant improvement of synaptic structures.

Please see Figure 3 and S3 above (Pages 5-7)

Major comment #2:

In Figure 5A, neuronal expression of Tau[WT] and Tau[R406W] do not exhibit significant sleep defects (blue lines in graph) under control sleep conditions. However, authors claim within the text that Tau[WT] and Tau [R406W] animals exhibit "significant sleep reduction and fragmentation due to Tau expression." Authors should explain the conflict between the data and text in *elav-GAL4*> UAS-Tau animals?

We recognized our deficiency in clarity. In this figure we presented just comparison between treatment groups to show differences after sleep modulation, because the significant sleep behavior changes were already presented in Figure 1. We have included a supplementary Figure S4 to show that the flies utilized for this experiment also showed sleep disturbances with pan-neuronal Tau expression. The revised results are as follows:

For the following experiments expressing Tau pan-neuronally, we first analyzed the sleep pattern and confirmed that these flies presented sleep disturbances as shown in Figure 1. After 4 days of sleep monitoring (Figure S4A), flies expressing Tau showed significant reduction of average sleep (Figure S4B), increased sleep fragmentation (Figure S4C) and decreased sleep length (Figure S4D).

Figure S4: Pan neuronal expression of Tau causes sleep disturbances. (A) Sleep profiles of 2 DAE flies expressing either *UAS-CD8-GFP* (black), *UAS-hTau^{WT}* (light pink) or *UAS-hTau^{R406W}* (magenta) pan-neuronally using *elav-GAL4* driver. Flies were allowed to acclimate from 0-48 hours and sleep was measured from 48-144 hours. (B) Quantification of average sleep per 24 hours measured in minutes. (C) Quantification of total number of bouts in 4 days (40-144 hours). (D) Quantification of average sleep bout length in minutes. Data as mean \pm SD. n = 6-10, One-way ANOVA, * $p < 0.05$, ** $p < 0.01$.

Major comment #3:

In figure 9, I am not convinced with the interpretation of the data, the authors observe an increase in autophagosomes in Tau[R406W] overexpressing brains after sleep induction. The authors claim that sleep induction leads to increased autophagy initiation. However, the data not show a concurrent increase in the autolysosomes in these neurons as one would expect if this is true. Plus, the increase in autophagosomes present class I neurons in sub-significant. It would be a more appropriate interpretation to state that sleep deprivation and induction does not have any significant effect on autophagy in the severe Tau[R406W] model due to severity of neurodegeneration induced by mutant Tau overexpression.

We thank the reviewer for pointing this out. This comment is similar to above major comment #5 from reviewer 1. To address this, we performed additional experiments to increase the number of flies and updated the original figures 6-9, now updated to **Figures 8-10 and Figure S5**.

As addressed above, we want to clarify that the results presented are of a snapshot in time of dissection and that the GFP and mCherry fluorescence intensity plotted is the pooled fluorescence of all Atg8 positive organelles present in each neuron, i.e. each data point represents the summation of all Atg8 containing organelle within one neuron at the time of dissection. The scattered plot shows the distribution of the neuronal population that expresses Tau under sleep modulation.

Moreover, we have included an additional measurement of GFP/mCherry ratio within a single cell, which serves as a proxy of the autophagic flux in the cell: a higher GFP/mCherry ratio is the result of either higher GFP signal due to a dysfunction of the cellular organelles with acidic pH, or lower mCherry signal, due to an increase in degradation, or a combination of both.

It is important to note that autophagic flux is a dynamic process where the rate of initiation vs degradation differs from cell to cell and time to time. To characterize such dynamic cellular process, we employed multiple independent approaches, (1) biochemical analysis of endogenous protein processing (WB and IP), (2) immunostaining of protein localization and expression, and (3) fluorescent reporter-based analyses. Based on the combined results of these approaches, we formulated our conclusions and presented our working models (Figure 11).

The revised results were the following:

Figure 8: Pan-neuronal Tau expression displays impaired autophagic flux.

Compared to the lacZ control group, Tau^{WT} and Tau^{R406W} expressing neurons showed a significant increase in Quadrant I, autophagosome accumulation, and a significant reduction in Quadrant II, fusion of the autophagosome with the lysosome (**Figure 8F**). Moreover, the significant increase in GFP/mCherry ratio in Tau expression neurons suggests a higher pH level and a dysfunction of the cellular organelles with acidic pH (**Figure 8G**). Together, these results reveal a shift in Atg8a puncta distribution in Tau pathology, and further suggest a potential block in autophagy flux at the step of fusion of autophagosome and lysosome (**Figure 8H**). The finding of reduced fusion and a consequent increase in autophagosome accumulation and an increase in GFP/mCherry ratio, would be consistent with Tau^{WT} or Tau^{R406W} expression-induced impairment of autophagic flux.

Figure 9: Sleep modulation promotes autophagic flux in lacZ.

Quantitative confocal imaging analysis of Atg8a puncta in the neuronal cell body region in the midbrain (**Figure 9C**), showed significant shift in puncta distribution under sleep modulation (**Figure 9D-9E**). Specifically, sleep deprivation caused a significant drop in Quadrant II, but no change in Quadrant III, while sleep induction caused a remarkable drop in Quadrant II and a concomitant increase

in Quadrant III, suggesting a significant reduction in puncta in the fusion state, and a significant increase in puncta undergoing lysosomal degradation, (**Figure 9E**). Moreover, there was significant increase in GFP/mCherry ratio after sleep deprivation and induction (**Figure 9F**). An increase in ratio could be due to either increase in GFP due to increased autophagosome accumulation or a decrease in mCherry due to increased degradation. These results suggest that under normal (non-pathological) conditions, the effects of sleep deprivation on autophagy are minimal, while sleep induction appears to promote autophagic flux and may facilitate lysosomal degradation (**Figure 9G**).

Figure 10: Sleep induction modulates autophagic flux in TauWT expression flies.

Imaging analysis of the Atg8a dual reporter expressed in the brain (**Figure 10C**) showed a significant increase in autophagosome puncta (Quadrant I) after sleep deprivation, accompanied by a decrease in Quadrant II. Moreover, after sleep induction a decrease in Quadrant II, with a concomitant increase in Quadrant III, was observed (**Figure 10D-10E**). In addition, the GFP/mCherry ratio was increased after sleep modulation (**Figure 10F**). These results suggest that under moderate pathology sleep deprivation leads to increased autophagosome accumulation and decreased fusion with the lysosome, while under sleep induction an increase in fusion and degradation occur (**Figure 10G**).

Figure S5: Sleep induction modulates autophagic flux in TauR406W expression flies.

However, analysis of autophagy flux using the Atg8a reporter revealed differences between the two pathological states (**Figure S5C**). Specifically, no significant changes were observed after sleep deprivation, while after sleep induction, we observed a decrease in Quadrant II (autolysosome) (**Figure S5D-S5E**) as well as an increase in GFP/mCherry ratio after sleep modulation (**Figure S5F**). These results suggest that under severe Tauopathy pathology, sleep modulation had a limited effect on autophagic flux, due to the system being severely degenerated. Although we observe minor changes after induction, there is high variability in the results (**Figure S5G**). Taken together, these results suggest that under moderate Tau pathology, sleep modulation promotes autophagic flux.

After further analysis we can conclude that subjecting Tau^{R406W} flies co-expressing the autophagy reporter to sleep modulation, there is high variability and no significant changes observed. These could be due to having a severe model, as well as stress response of the combined genotype, sleep modulation and Atg8a induced manipulation. Therefore, we have moved this figure to supplementary materials.

Please see Figure 8-10 and S5 above (Pages 16-23)

References

1. Ma, X., et al., *Nicotinamide mononucleotide adenylyltransferase uses its NAD⁺ substrate-binding site to chaperone phosphorylated Tau*. eLife, 2020. **9**.
2. Dissel, S., et al., *Sleep Restores Behavioral Plasticity to Drosophila Mutants*. Current Biology, 2015. **25**(10): p. 1270-1281.
3. Lopez-Ortega, E., R. Ruiz, and L. Tabares, *CSP α , a Molecular Co-chaperone Essential for Short and Long-Term Synaptic Maintenance*. Frontiers in Neuroscience, 2017. **11**.

August 27, 2024

RE: Life Science Alliance Manuscript #LSA-2024-02681-TR

Dr. Rong Grace Zhai
University of Chicago
Neurology
5841 S Maryland Ave
MC 2030, J310
Chicago, IL 60637

Dear Dr. Zhai,

Thank you for submitting your revised manuscript entitled "Regulation of proteostasis by sleep through autophagy in *Drosophila* models of Alzheimer's Disease". We would be happy to publish your paper in Life Science Alliance pending final revisions necessary to meet our formatting guidelines.

- please be sure that the authorship listing and order is correct
- please add ORCID ID for the corresponding author -- you should have received instructions on how to do so
- please add the Twitter handle of your host institute/organization as well as your own or/and one of the authors in our system
- please add an Author Contributions section to your main manuscript text
- please add a Conflict of Interest statement to your main manuscript text
- please move information such as Data Availability, Acknowledgements, Author contributions, and Conflict of interest after the material and methods section and before the references
- we encourage you to revise the figure legend for Figure S5 such that the figure panels are introduced in alphabetical order
- it looks like there are no tables provided in your manuscript. Please correct the label of the section for the legends
- you may want to consider uploading Figure 11 as a Graphical Abstract, rather than as a figure, but this is up to you

FIGURE CHECKS:

-there appears to be a splice after the 4th lane in Figure 7F. If this is correct, please add a vertical black line at the splice through the length of the gel and mention in the figure legend that the black line indicates a splice in the blot

A. FINAL FILES:

B. MANUSCRIPT ORGANIZATION AND FORMATTING:

Thank you for your attention to these final processing requirements. Please revise and format the manuscript and upload materials within 5 days.

Sincerely,

Reviewer #1 (Comments to the Authors (Required)):

The rebuttal has addressed the concerns I raised so the manuscript is OK to accept now.

Reviewer #2 (Comments to the Authors (Required)):

This manuscript performs a thorough phenotypic analysis on the role of sleep on the Tau neurotoxicity in *Drosophila*. In addition, the authors describe a mechanism by which sleep may function in regulating Tau pathogenesis in the adult brain. Moreover, this paper provides a model system to properly dissect the relationship between sleep abnormalities and neurodegenerative disease. The authors did a thorough job addressing the concerns in the initial manuscript in the revisions and present additional data to support their conclusions. I recommend this manuscript for publication in Life Science Alliance.

August 29, 2024

RE: Life Science Alliance Manuscript #LSA-2024-02681-TRR

Dr. Rong Grace Zhai
University of Chicago
Department of Neurology
5841 S Maryland AVE
MC2030, J310
Chicago, IL 60637

Dear Dr. Zhai,

Thank you for submitting your Research Article entitled "Regulation of proteostasis by sleep through autophagy in Drosophila models of Alzheimer's Disease". It is a pleasure to let you know that your manuscript is now accepted for publication in Life Science Alliance. Congratulations on this interesting work.

DISTRIBUTION OF MATERIALS:

Again, congratulations on a very nice paper. I hope you found the review process to be constructive and are pleased with how the manuscript was handled editorially. We look forward to future exciting submissions from your lab.

Sincerely,
